# Mitochondrial KMT9 methylates DLAT to control pyruvate dehydrogenase activity and prostate cancer growth

Yanhan Jia[1,2], Sheng Wang ®[1], Sylvia Urban[1], Judith M. Müller ®[1], Manuela Sum[1], Qing Wang[3], Helena Bauer[1], Uwe Schulte ®[4,5], Heike Rampelt ®[5,6], Nikolaus Pfanner ®[5,6], Katrin M. Schüle ®[7], Axel Imhof ®[8], Ignasi Forné ®[8], Christopher Berlin ®[9], August Sigle[1], Christian Gratzke[1], Holger Greschik[1], Eric Metzger ®[1,2] ✉ & Roland Schüle ®[1,2,5] ✉

Prostate cancer (PCa) growth depends on de novo lipogenesis controlled by the mitochondrial pyruvate dehydrogenase complex (PDC). In this study, we identify lysine methyltransferase (KMT)9 as a regulator of PDC activity. KMT9 is localized in mitochondria of PCa cells, but not in mitochondria of other tumor cell types. Mitochondrial KMT9 regulates PDC activity by mono-methylation of its subunit dihydrolipoamide transacetylase (DLAT) at lysine 596. Depletion of KMT9 compromises PDC activity, de novo lipogenesis, and PCa cell proliferation, both in vitro and in a PCa mouse model. Finally, in human patients, levels of mitochondrial KMT9 and DLAT K596me1 correlate with Gleason grade. Together, we present a mechanism of PDC regulation and an example of a histone methyltransferase with nuclear and mitochondrial functions. The dependency of PCa cells on mitochondrial KMT9 allows to develop therapeutic strategies to selectively fight PCa.

Mitochondria play a central role in metabolism, aerobic respiration, and redox homeostasis[1]. Mitochondrial metabolism has emerged as a positive regulator of tumorigenesis and cancer progression[1,2]. While cancer cells typically depend on glycolysis for energy metabolism and biosynthesis under aerobic conditions, a phenomenon known as Warburg effect[3], examples of metabolic reprogramming towards an oxidative cancer metabolism have been documented[4,5]. Prostate cancer (PCa) cells exhibit untypical requirements and are characterized by particular metabolic dependencies[4,5]. Non-transformed prostate epithelial cells operate a truncated tricarboxylic acid (TCA) cycle to generate high amounts of citrate, which is secreted into the seminal lumen. In contrast, PCa cells reprogram mitochondrial function to restore a fully operational TCA cycle and enhance mitochondrial metabolism. These metabolic adaptations contribute to cancer cell anabolism by increasing, for example, de novo lipogenesis, which supports tumor growth[2,4]. Accordingly, targeting mitochondrial metabolism has emerged as a potential therapeutic strategy for the treatment of PCa[6,7].

The pyruvate dehydrogenase complex (PDC) catalyzes the oxidative decarboxylation of pyruvate to acetyl coenzyme A (acetyl-CoA), thereby connecting glycolysis with the TCA cycle and controlling mitochondrial metabolism. PDC is composed of multiple copies of

[1]Klinik für Urologie und Zentrale Klinische Forschung, Klinikum der Universität Freiburg, Freiburg, Germany. [2]German Cancer Consortium (DKTK), Partner site Freiburg, a partnership between DKFZ and Medical Center - University of Freiburg, Freiburg, Germany. [3]Complete Omics Inc., Baltimore, MD, USA. [4]Institute of Physiology II, Faculty of Medicine, University of Freiburg, Freiburg, Germany. [5]CIBSS Centre of Biological Signalling Studies, University of Freiburg, Freiburg, Germany. [6]Institute of Biochemistry and Molecular Biology, Faculty of Medicine, University of Freiburg, Freiburg, Germany. [7]Institute of Experimental and Clinical Pharmacology and Toxicology, Faculty of Medicine, University of Freiburg, Freiburg, Germany. [8]Institute Biomedical Center, Protein Analysis Unit, Faculty of Medicine, Ludwig-Maximilians-Universität München, Planegg-Martinsried, Germany. [9]Department of General and Visceral Surgery, Faculty of Medicine, University of Freiburg, Freiburg, Germany. ✉e-mail: eric.metzger@uniklinik-freiburg.de; roland.schuele@uniklinik-freiburg.de

three enzymatic subunits: pyruvate dehydrogenase (PDHA1/PDHB, E1 subunit), DLAT (E2 subunit), and dihydrolipoamide dehydrogenase (DLD, E3 subunit) as well as an additional structural subunit, pyruvate dehydrogenase complex component X (PDHX), mediating E2/E3 subunit interactions[8]. The enzymatic activity of PDC is regulated by post-translational modifications. Phosphorylation of PDHA1 by pyruvate dehydrogenase kinase (PDK) at serine (S) 232, S293, or S300 is known to inhibit PDC activity[9]. Besides, acetylation of PDHA1[10,11], phosphorylation of DLD[12], lipoylation of DLAT[13,14], and modification with O-linked β-N-acetylglucosamine (O-GlcNAcylation) of both DLAT and PDHX[15] are required for full PDC activity[16]. In PCa cells, mitochondrial PDC plays a pivotal role in supporting de novo lipogenesis. Acetyl-CoA produced by PDC is converted to citrate, which is transported out of mitochondria to the cytosol and converted to acetyl-CoA by ATP citrate lyase (ACLY). Acetyl-CoA is then used for the production of lipid building blocks during de novo lipogenesis to support tumor cell proliferation[2].

KMT9 was recently described as a histone methyltransferase monomethylating histone H4 at lysine 12 (H4K12)[17]. The enzyme is an obligate heterodimer composed of KMT9α, which harbors the methyltransferase activity, and KMT9β serving a structural role[17–19]. In addition to monomethylation of H4K12, KMT9 has been observed to methylate other target proteins[17–22]. KMT9 expression is increased in several types of cancer including prostate, lung, and colon carcinoma[17,23,24]. In these cancer types, KMT9 controls the transcription of nuclear target genes involved in cell cycle regulation, thereby promoting tumor cell proliferation[17,23,24].

In this study, we uncover an important function of mitochondrial KMT9 in PCa. KMT9 is only detected in mitochondria of normal prostate and PCa cells, whereas mitochondria of other tumor cell types are devoid of KMT9. We show that mitochondrial KMT9 regulates the activity of PDC by monomethylation of its E2 subunit DLAT at K596, thereby controlling de novo lipogenesis and cancer cell proliferation. Compromised PDC activity upon KMT9α depletion can be rescued with exogenous KMT9α fused to a mitochondrial targeting sequence (MTS) but not with KMT9α harboring a nuclear localization signal (NLS). Similarly, abrogated PDC activity upon DLAT depletion can be rescued with exogenous wild-type DLAT but not a methylation-defective mutant DLAT (K596R). Compared to compromised PDC activity, the full rescue of impaired prostate tumor cell proliferation requires expression of both MTS-KMT9α and NLS-KMT9α, which is in accordance with our previous characterization of KMT9 functions in the nucleus[17]. Importantly, KMT9 methylates DLAT and controls de novo lipogenesis in a PCa mouse model. Furthermore, we uncover that in human patients DLAT methylation correlates with Gleason grade. Together, we identify KMT9 as an essential regulator of PCa mitochondrial function contributing to the understanding of specific requirements of prostate carcinoma. Our data suggest that targeting of mitochondrial KMT9, possibly in combination with drugs targeting metabolic regulators in other cell compartments and/or inhibitors of nuclear KMT9 function, can be exploited as a potential therapeutic approach for the treatment of PCa.

## Results

### KMT9 is localized in mitochondria of normal prostate and PCa cells

We analyzed the subcellular localization of KMT9α and its heterodimer partner KMT9β in several PCa cell lines by cell fractionation and immunofluorescence staining. In accordance with previous data[17], we observed KMT9 localization in the nucleus and the cytoplasm (Fig. 1a, b, and Supplementary Fig. 1a). Interestingly, in all PCa cell lines tested, we found a fraction of KMT9 present in mitochondria (Fig. 1a, b, and Supplementary Fig. 1a). The mitochondrial localization of KMT9 in PCa cells was validated by proteinase K protection assays showing that KMT9 is localized in the mitochondrial matrix (Fig. 1c, and

Supplementary Fig. 1b–d). To further verify mitochondrial presence, we used an *in organello* import assay and observed that KMT9α and KMT9β were imported into mitochondria isolated from PCa cell lines (Fig. 1d and Supplementary Fig. 1e). We also detected KMT9 in mitochondria of normal prostate epithelial cell lines PNT1A and PNT2, albeit at lower levels compared to PCa cells (Fig. 1e). Strikingly, mitochondrial KMT9 was not observed in an extensive panel of 74 cell lines representing 28 different tumor entities including liver, lung, pancreas, colon, and breast cancer (Fig. 1e and Supplementary Fig. 1f). In accordance with the observations in cell lines, we detected mitochondrial KMT9α in mouse primary prostate epithelial cells (Supplementary Fig. 1g) and mouse prostate tissue but not in liver, spleen, kidney, heart, brain or bladder (Supplementary Fig. 1h), while KMT9β was observed in mitochondria of prostate, spleen, brain and bladder (Supplementary Fig. 1h). Since KMT9α requires dimerization with KMT9β for catalytic activity, active KMT9 can only form in mitochondria of mouse prostate tissue. These data suggest that mitochondrial localization of KMT9 may be a distinct feature of prostate cells.

### KMT9 regulates PDC activity

To investigate potential functions of KMT9 in mitochondria, we first searched for interaction partners of KMT9 and performed immunoprecipitation coupled with liquid chromatography-tandem mass spectrometry (LC-MS/MS). Intersection of the proteins identified by LC-MS/MS with the human MitoCarta 3.0 inventory comprising proteins with mitochondrial localization[25] suggested an interaction of KMT9 with several PDC subunits (DLAT, PDHA1, PDHB, PDHX) (Fig. 1f, Supplementary Data 1) hinting at a functional connection between KMT9 and PDC. By co-immunoprecipitation, we observed an interaction between endogenous, mitochondrial KMT9 and DLAT, a key catalytic component of PDC (Fig. 1g). Given the association between KMT9 and DLAT, we next sought to investigate potential metabolic consequences of KMT9 depletion. To this end, we profiled the metabolome of PC-3M cells treated with a control siRNA (siCtrl) or a siRNA directed against the *KMT9α* coding region (siKMT9α #2) by ultra-performance liquid chromatography-tandem mass spectrometry (UPLC-MS/MS) (Fig. 1h). Notably, the metabolomic analysis showed that KMT9α depletion led to an increase in the levels of pyruvate, NAD⁺, and lactate, as well as a decrease in the level of acetyl-CoA (Fig. 1i–l, Supplementary Data 2). These data suggested impaired PDC activity as a consequence of KMT9α depletion in PCa cells (Fig. 1m).

Consequently, we examined the impact of KMT9α depletion on the activity of endogenous PDC. KMT9α depletion in PCa cells using a siRNA directed against the 3′ untranslated region (siKMT9α #1) or siKMT9α #2 led to a strong decrease in PDC activity (Fig. 1n and Supplementary Fig. 2a, b). Ectopically expressed KMT9α rescued the impaired PDC activity caused by KMT9α depletion, whereas a catalytically inactive mutant KMT9α (N122A)[17] failed to rescue. Furthermore, decreased PDC activity was restored by MTS-KMT9α but neither by NLS-KMT9α nor MTS-KMT9α (N122A) (Fig. 1n and Supplementary Fig. 2b). Next, we immunoprecipitated PDC from PC-3M cells treated with siCtrl or siKMT9α and assayed PDC activity in vitro. In vitro supplementation of immunocaptured PDC with recombinant, purified KMT9 heterodimer (rKMT9α/KMT9β), but not with enzymatically inactive rKMT9α (N122A)/KMT9β, restored full PDC activity (Fig. 1o). We also analyzed mitochondrial gene expression by quantitative RT-PCR analysis, but detected no or only marginal changes (Supplementary Fig. 2c), which are unlikely to explain impaired PDC activity. Together, these findings demonstrated a critical role of catalytically active, mitochondrial KMT9 in the regulation of PDC activity.

### KMT9 monomethylates DLAT at lysine 596

Next, we investigated how KMT9 modulates PDC activity. We hypothesized that a PDC subunit interacting with KMT9 might be

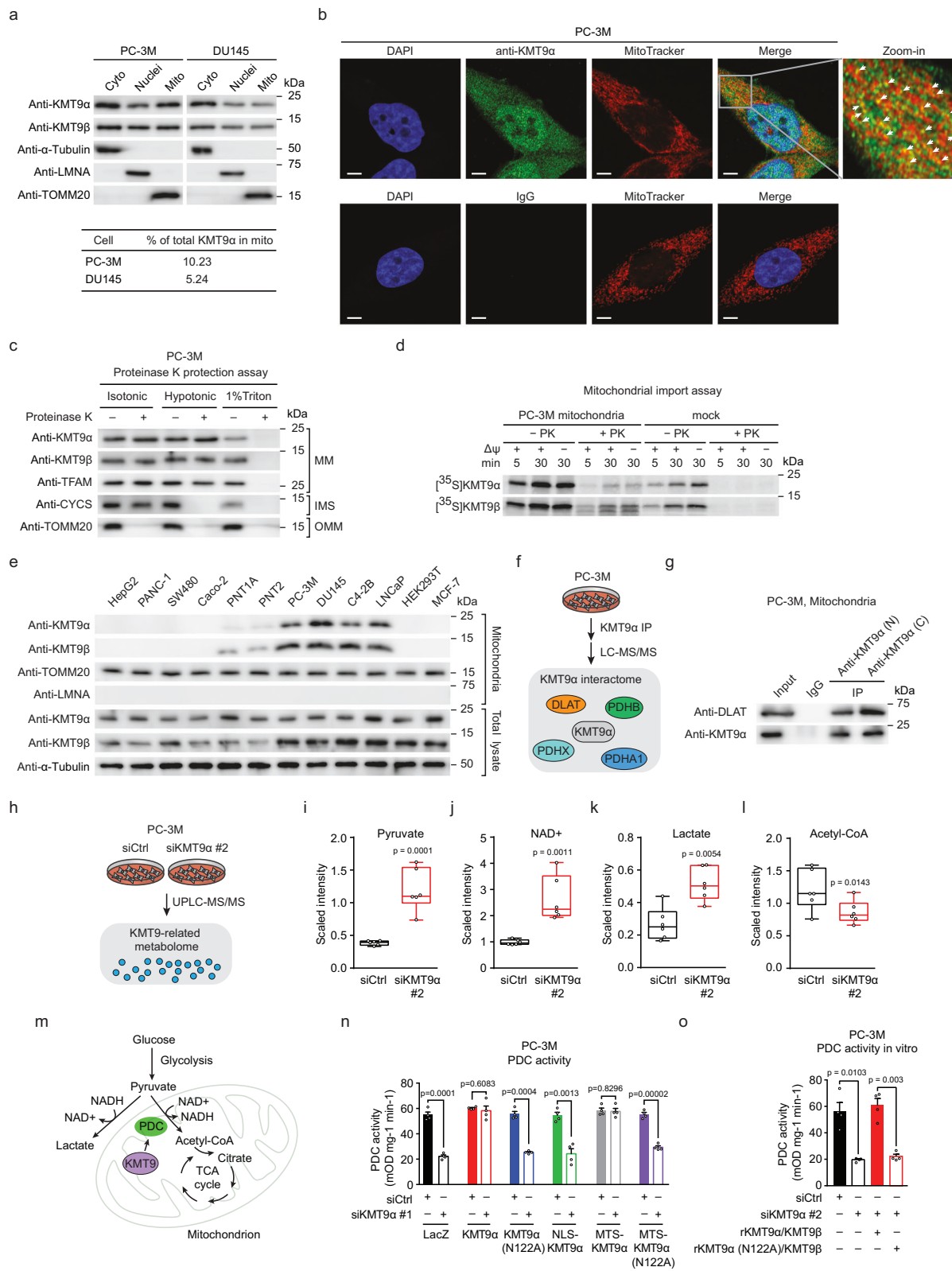

methylated and consequently probed PDC, immunocaptured from mitochondria, with an anti-pan-methyl lysine antibody. Our assays revealed methylation of a protein with the molecular weight corresponding to that of DLAT, which was abolished upon KMT9α depletion (Fig. 2a). To verify that DLAT was indeed methylated by KMT9 and to identify the methylated amino acid residue, we analyzed DLAT immunopurified from mitochondria with targeted mass spectrometry

and identified monomethylation of lysine (K) 596 (DLAT K596me1) (Supplementary Fig. 3a). We next generated an antibody selectively recognizing DLAT K596me1 (Supplementary Fig. 3b–d). Using this antibody, we confirmed that KMT9α depletion reduced DLAT K596me1 levels without affecting DLAT expression (Fig. 2b and Supplementary Fig. 3e). DLAT methylation in KMT9α-depleted cells could be restored by expression of exogenous KMT9α but not the

**Fig. 1 | KMT9 is localized in mitochondria and regulates PDC activity in PCa cells. a** Subcellular fractionation analysis of PC-3M or DU145 cells with quantification of mitochondrial KMT9α percentage. **b** Immunofluorescence staining of PC-3M cells showing KMT9α, mitochondria (MitoTracker), and nuclei (DAPI). Scale bar: 5 µm. **c** Proteinase K protection assay for mitochondria isolated from PC-3M cells. TFAM (mitochondrial matrix, MM), CYCS (intermembrane space, IMS), and TOMM20 (outer mitochondrial membrane, OMM) served as controls for compartmental digestion. **d** Import of [35S]-labeled KMT9α and KMT9β into mitochondria isolated from PC-3M cells. Δψ, membrane potential; PK, proteinase K. **e** Presence of KMT9α and KMT9β in mitochondrial extracts of diverse cancer and non-cancer cell lines was revealed by Western blot analysis using the indicated antibodies. **f** Schematic illustration of KMT9α interactome in PC-3M cells. **g** Co-immunoprecipitation using antibodies against either the N-terminus (N) or the C-terminus (C) of KMT9α showing the interaction between DLAT and KMT9α in mitochondrial lysates. (Input: 5% of total extract). **h** Schematic illustration of global metabolomics analyses in siCtrl or siKMT9α-treated PC-3M cells. **i–l** Relative abundance of pyruvate (**i**), NAD$^+$ (**j**), lactate (**k**), and acetyl-CoA (**l**) identified by global metabolomics analyses in siCtrl or siKMT9α-treated PC-3M cells. **m** Schematic showing pyruvate conversion to acetyl-CoA by PDC in mitochondria. **n** PDC activity in PC-3M cells expressing different KMT9α variants with or without endogenous KMT9α depletion. **o** Restoration of PDC activity in vitro. Extracts of PC-3M cells transfected with siCtrl or siKMT9α were supplemented with recombinant wildtype KMT9 heterodimer (rKMT9α/KMT9β) or catalytically inactive heterodimer (rKMT9α (N122A)/KMT9β). **i–l, n, o** Data are shown as box plots (**i–l** n = 6 biological replicates; center line indicates median, box bounds show 25th and 75th percentiles, whiskers represent minimum and maximum values within 1.5 times the interquartile range) or mean + SD (**n, o** n = 4 biological replicates). Statistical significance was determined by two-sided Welch's two-sample t-test with false discovery rate (FDR) correction (**i–l**) or two-sided Student's t-test (**n, o**). (**a–e, g, n, o**). All experiments were independently repeated at least three times with similar results. Source data are provided as a Source Data file.

catalytically inactive mutant KMT9α (N122A) (Fig. 2b). Importantly, incubation of PDC immunocaptured from PCa cells with rKMT9α/KMT9β, but not rKMT9α (N122A)/KMT9β, strongly increased DLAT K596me1 levels in vitro (Fig. 2c and Supplementary Fig. 3f). These data demonstrated that KMT9 monomethylates DLAT at K596. Of note, we detected robust levels of KMT9 and DLAT K596me1 in mitochondria of human PCa specimens compared to lower levels in tumor-adjacent normal prostate tissue (Fig. 2d). In contrast, we neither observed methylated DLAT in mitochondria of human healthy tissues such as mesenchyme or colon nor in human bladder cancer tissue (Fig. 2d). These data further corroborated the prostate-selective mitochondrial localization of KMT9 and DLAT K596me1 observed in human cell lines.

Since KMT9 and PDC also reside in the nucleus, we investigated by Western blot whether nuclear DLAT was methylated by KMT9. We exclusively detected DLAT K596me1 in the mitochondrial but not in the nuclear or cytosolic fraction of PCa cells (Fig. 2e). The exclusive mitochondrial localization of DLAT K596me1 was verified by confocal laser scan microscopy of DU145 and PC-3M prostate cancer cells (Supplementary Fig. 3g). Consistently, co-immunoprecipitation assays using nuclear fractions of DU145 and PC-3M cells transfected with NLS-KMT9α showed that neither endogenous KMT9α nor ectopically expressed NLS-KMT9α interacted with nuclear DLAT, explaining why nuclear KMT9 fails to methylate DLAT (Supplementary Fig. 4a–d). In accordance with these results, neither depletion of KMT9α nor ectopic expression of NLS-KMT9α affected acetylation of H3K9 (H3K9ac), a known modification requiring nuclear PDC-derived acetyl-CoA[2] (Supplementary Fig. 4e, f). Of note, in vitro methylation assays demonstrated that recombinant NLS-KMT9α/KMT9β retained the ability to methylate DLAT at K596 (Supplementary Fig. 4g, h. Therefore, the absence of methylated nuclear DLAT does not result from impaired catalytic activity of nuclear KMT9. Together, our data provide multiple lines of evidence that KMT9-mediated DLAT K596 methylation is restricted to mitochondria and does not occur in the nuclear compartment of PCa cells.

## DLAT methylation by KMT9 is required for PDC activity
To investigate whether methylation of DLAT at K596 affected PDC activity, we ectopically expressed wildtype DLAT, mutant DLAT (K596R), which cannot be methylated by KMT9, or a control mutant DLAT (K547R) upon depletion of endogenous DLAT using siRNA directed against the 3′ untranslated region (siDLAT #1) (Fig. 2f and Supplementary Fig. 4i). As expected, the K596R mutation abolished DLAT methylation, whereas methylation of the control mutant DLAT (K547R) was unchanged (Fig. 2f and Supplementary Fig. 4i). Importantly, impaired PDC activity upon depletion of endogenous DLAT was rescued by expression of exogenous DLAT or the control mutant DLAT (K547R), but not the methylation-defective mutant DLAT (K596R) (Fig. 2g and Supplementary Fig. 4j).

As additional controls, we investigated whether DLAT methylation by KMT9 indirectly affected PDC function. KMT9α knockdown did not significantly affect the expression levels of PDC subunits (Supplementary Fig. 4k). Furthermore, we checked for increased PDHA1 phosphorylation at S232, S293, and S300, which is known to inhibit PDC activity[9] but observed only a small decrease in phospho-S293 and no significant changes for S232 and S300. However, upon DLAT knockdown, we noted a decrease in phosphorylation of S300, whereas phospho-S232 and -S293 levels were unchanged (Supplementary Fig. 4l). Finally, we checked whether the DLAT K596R mutation affected DLAT lipoylation, an indispensable posttranslational modification for PDC activity[13,14]. We observed that exogenous DLAT and DLAT (K596R) were lipoylated to the same extent in DLAT-depleted cells (Supplementary Fig. 4l), which shows that the reduced PDC activity upon DLAT mutation is due to the loss of methylation at DLAT K596 rather than an indirect effect on lipoylation.

## DLAT methylation contributes to PCa cell proliferation
Since reduced DLAT K596 methylation affected PDC activity, we next investigated potential effects on PCa cell proliferation. DLAT depletion with siDLAT #1 or siDLAT #2 impaired proliferation of PCa cell lines (Fig. 3a–c and Supplementary Fig. 5a–c), which is in accordance with previous results obtained upon knockdown of the PDC component PDHA1[2]. Proliferation defects could be rescued by expression of exogenous wild-type DLAT but not the methylation-defective mutant DLAT (K596R) (Fig. 3a–c and Supplementary Fig. 5d–i). DLAT knockdown also suppressed proliferation of other cancer cell types including SW480 (colorectal cancer) (Supplementary Fig. 6a) and HT1376 (bladder cancer cells) (Supplementary Fig. 6b). However, contrary to PCa cells, proliferation defects of SW480 and HT1376 cells could be rescued with exogenous DLAT K596R (Supplementary Fig. 6c–h) showing that DLAT methylation at K596 is dispensable for their growth.

Furthermore, we observed decreased PCa cell proliferation upon KMT9α depletion, which was fully restored by ectopic expression of KMT9α (Fig. 3d and Supplementary Fig. 7a, g). In comparison, exogenous MTS-KMT9α or NLS-KMT9α only partially restored the proliferation defect of PCa cells (Fig. 3e, f and Supplementary Fig. 7b, c, h, i). Importantly, the full proliferative capacity of PCa cells was only restored upon expression of both, MTS-KMT9α and NLS-KMT9α, whereas expression of the catalytically inactive mutants MST-KMT9α (N122A) and NLS-KMT9α (N122A) failed to rescue (Fig. 3g, h and Supplementary Fig. 7d, e, j, k). Knockdown efficiency and expression of exogenous KMT9α proteins was confirmed by Western blot (Fig. 3i and Supplementary Fig. 7f–l). These data are in full agreement with our previous study describing the functions of nuclear KMT9 in PCa cell proliferation[17]. In non-prostate cancer cells, expression of either KMT9α or NLS-KMT9α fully rescued the proliferation defect induced

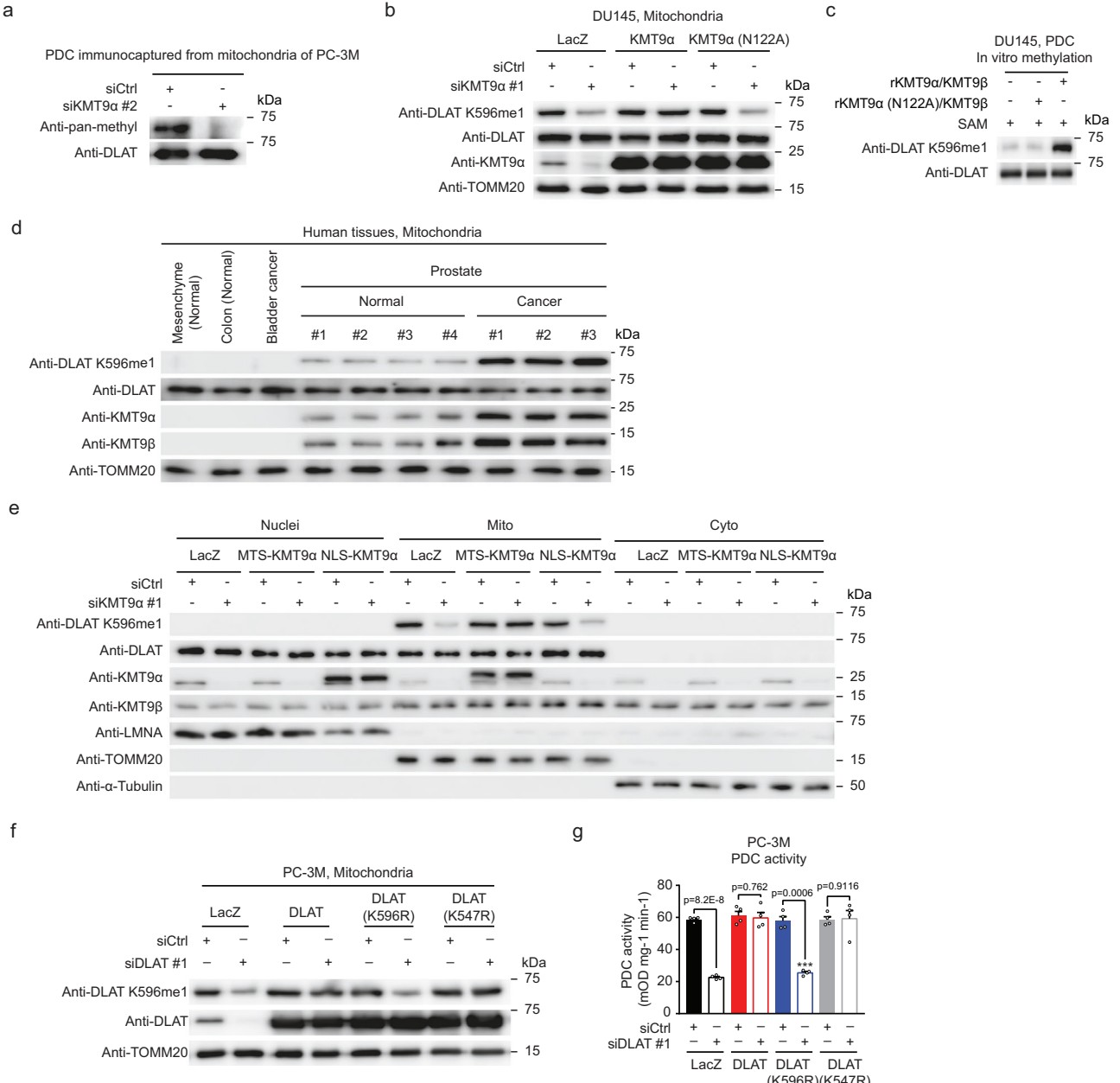

**Fig. 2 | Mitochondrial KMT9 regulates PDC activity by monomethylation of DLAT at K596. a** Western blot analysis of PDC immunocaptured from mitochondrial fractions of siCtrl- or siKMT9α-treated PC-3M cells using anti-pan-methyllysine antibody. **b** Detection of DLAT K596me1 in mitochondrial fractions of DU145 cells treated with siCtrl or siKMT9α and transfected with expression plasmid for LacZ, KMT9α, or catalytically inactive KMT9α (N122A) by Western blot. Membranes were decorated with the indicated antibodies. TOMM20 served as loading control. **c** In vitro methylation of DLAT. PDC immunocaptured from DU145 cells was incubated with recombinant (r) KMT9α/β or rKMT9α (N122A)/KMT9β and S-adenosyl methionine (SAM). Methylation reactions were analyzed by Western blot using anti-DLAT K596me1 or anti-DLAT antibody. **d** Detection of DLAT K596me1, KMT9α, and KMT9β in mitochondrial extracts of indicated human specimens by Western blot.

**e** Detection of DLAT K596me1 in nuclear, mitochondrial, or cytosolic fractions of DU145 cells transfected with siCtrl or siKMT9α and expression plasmid for LacZ, MTS-KMT9α, or NLS-KMT9α by Western blot. LMNA, TOMM20, and α-Tubulin served as controls for nuclear, mitochondrial, and cytosolic fractions, respectively. **f**, **g** Activity of PDC immunocaptured from extracts of PC-3M cells transfected with expression plasmid for LacZ, DLAT, DLAT (K596R), or DLAT (K547R) in combination with siCtrl or siDLAT. Knockdown efficiency and expression of exogenous DLAT proteins was verified by Western blot (**f**). Data represent mean + SD from 4 biological replicates. Statistical significance was determined by two-sided Student's *t*-test. (**a–g**) All experiments were independently repeated at least three times with similar results. Source data are provided as a Source Data file.

by KMT9α depletion, whereas MTS-KMT9α failed to do so, providing further evidence that mitochondrial KMT9 is not required (Supplementary Fig. 8a–j). Together, our results demonstrate an indispensable role of KMT9-mediated DLAT K596 methylation for PDH activity driving PCa cell proliferation (Fig. 3j). Furthermore, we provide an important example of a histone methyltransferase with dual functions in the nucleus and in mitochondria.

## KMT9-driven mitochondrial functions affect de novo lipogenesis

PDC-mediated mitochondrial respiration was previously reported to fuel PCa cell proliferation via the regulation of de novo lipogenesis[2]. Consequently, we assessed the impact of KMT9 loss on lipogenesis. Using ELISA-based quantification assays, we observed that depletion of KMT9α led to a significant decrease in the levels of both free fatty acids

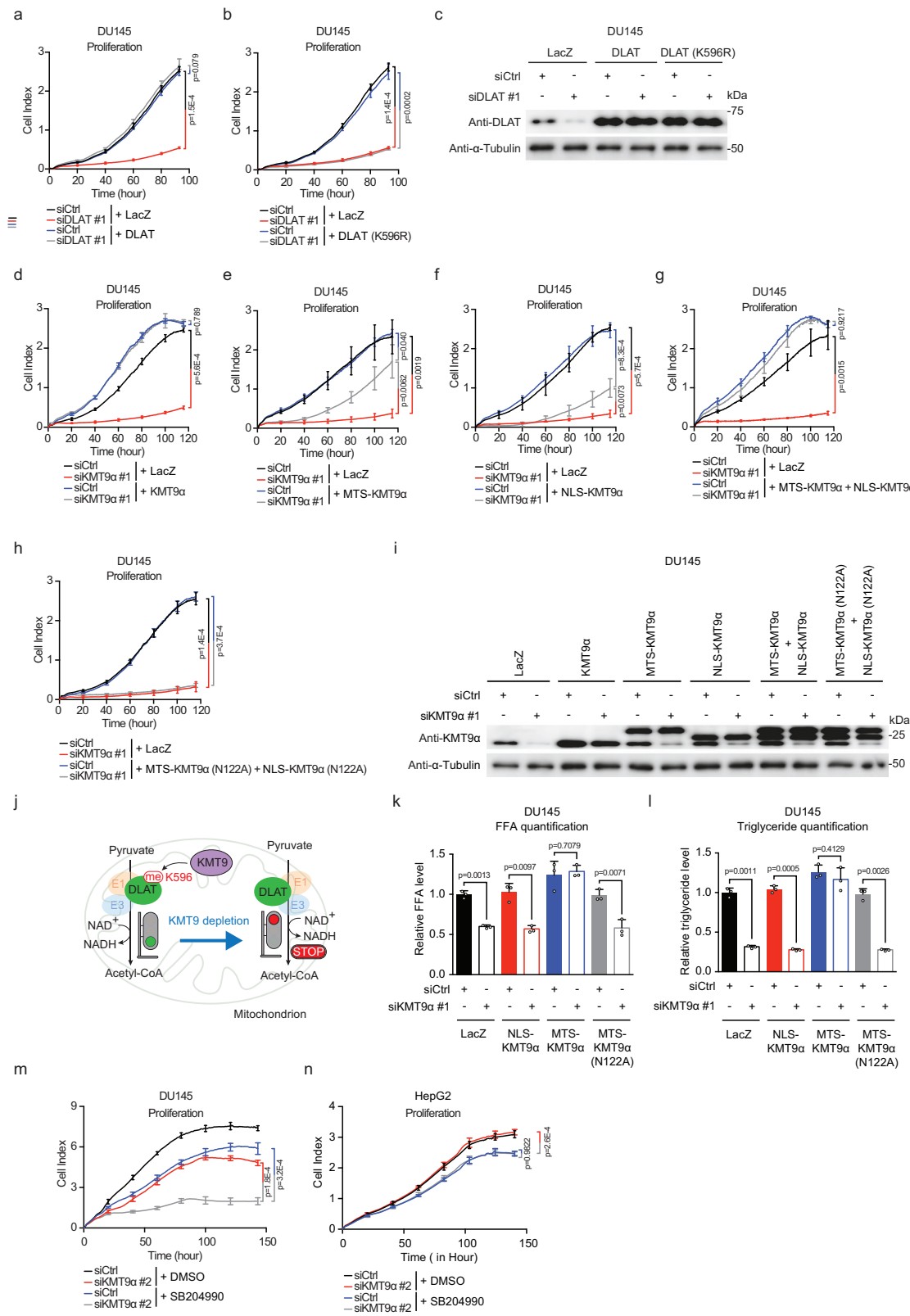

and triglycerides in PCa cells (Fig. 3k, l and Supplementary Fig. 9a, b) suggesting a defect in lipogenesis. In contrast, KMT9α depletion in non-prostate cancer cells did not affect the levels of free fatty acids and triglycerides (Supplementary Fig. 9c, d). Importantly, exogenous expression of MTS-KMT9α, but not NLS-KMT9α, completely restored the reduced levels of free fatty acids and triglycerides in PCa cells (Fig. 3k, l and Supplementary Fig. 9e–j) indicating that the

mitochondrial but not the nuclear pool of KMT9α is indispensable for the regulation of lipogenesis. Of note, exogenous expression of the enzymatically inactive mutant MTS-KMT9α (N122A) failed to rescue the phenotype (Fig. 3k, l and Supplementary Fig. 9e-j) demonstrating that the role of mitochondrial KMT9α in the regulation of de novo lipogenesis relies on its methyltransferase activity. Supplementation with the fatty acid oleate partially restored impaired proliferation

**Fig. 3 | Mitochondrial KMT9 fuels PCa cell proliferation via regulation of de novo lipogenesis. a–c** Proliferation of DU145 cells transfected with siCtrl or siDLAT in combination with expression plasmid for LacZ (**a**, **b**), DLAT (**a**), or DLAT (K596R) (**b**) as indicated. Knockdown efficiency and expression of exogenous DLAT proteins was verified by Western blot (**c**). α-Tubulin served as loading control.
**d–i** Proliferation of DU145 cells transfected with siCtrl or siKMT9α in combination with an expression plasmid for LacZ (**d–h**), KMT9α (**d**), MTS-KMT9α (**e**, **g**), NLS-KMT9α (**f**, **g**), or MTS-KMT9α (N122A), and NLS-KMT9α (N122A) (**h**). Knockdown efficiency and expression of exogenous KMT9α proteins was verified by Western blot (**i**). α-Tubulin served as loading control. **j** Proposed mechanism for KMT9-mediated control of mitochondrial functions in PCa cells. KMT9 monomethylates DLAT at K596, which is required for PDC activity and mitochondrial metabolism.

KMT9 depletion abolishes DLAT K596 monomethylation, thereby impairing PDC activity and mitochondrial metabolism in PCa cells. **k, l** Free fatty acid (FFA) (**k**) and triglyceride (**l**) levels in DU145 cells transfected with siCtrl or siKMT9α in combination with expression plasmid for LacZ, NLS-KMT9α, MTS-KMT9α, or MTS-KMT9α (N122A). **m, n** Proliferation of DU145 (**m**) or HepG2 (**n**) cells transfected with sub-optimal concentrations of siCtrl or siKMT9α in the presence or absence of 100 μM SB204990 to detect potential effects of co-treatment. **a, b, d–h, k–n**, Data are presented as mean ± SD (**a, b, d–h, m, n**, $n = 4$ biological replicates) or mean + SD (**k, l**, $n = 3$ biological replicates). Statistical significance was determined by a two-sided Student's t-test (**a, b, d–h, k–n**). All experiments were independently repeated at least three times with similar results. Source data are provided as a Source Data file.

caused by KMT9α depletion (Supplementary Fig. 9k, l), which is in accordance with the partial rescue by MTS-KMT9α (Fig. 3e and Supplementary Fig. 7b, h).

Finally, we examined whether targeting de novo lipogenesis and concomitant KMT9 depletion would allow the efficient suppression of PCa cell proliferation at lower individual treatment doses. To this end, we utilized two well-documented inhibitors (SB204990[2] and NDI-091143[26]) of ACLY, a key enzyme that controls de novo lipogenesis, in combination with KMT9α depletion. When used at suboptimal doses, siKMT9 and ACLY inhibitors only slightly affected cell proliferation (Fig. 3m and Supplementary Fig. 10a–c). In contrast, KMT9α depletion in combination with either of the two ACLY inhibitors led to robust inhibition of PCa cell proliferation (Fig. 3m and Supplementary Fig. 10a–c), which was not observed in non-PCa cells (Fig. 3n and Supplementary Fig. 10d–g) devoid of mitochondrial KMT9. Furthermore, the partial rescue of cell proliferation achieved by ectopic expression of MTS-KMT9α in KMT9α-depleted PCa cells could be reversed by treatment with lipogenesis inhibitor (Supplementary Fig. 10h, i), whereas the rescue by NLS-KMT9α was insensitive to lipogenesis inhibition (Supplementary Fig. 10j, k). Taken together, these results provide strong evidence that KMT9-mediated mitochondrial respiration fuels PCa cells via modulating de novo lipogenesis.

## KMT9 controls PDC activity, de novo lipogenesis, and tumor growth in mice

To explore whether KMT9 affected DLAT methylation and activity in vivo, we used the well-established *Nkx3.1-Cre-ER*[T2(Tg/Tg)]/*Pten*[fl/fl]/*Smad4*[fl/fl] PCa mouse model[27,28]. The combinatorial deletion of the tumor suppressors *Phosphatase and tensin homolog* (*Pten*) and *SMAD family member 4* (*Smad4*) in prostate luminal cells (hereafter termed *Pten/ Smad4* KO) induces highly aggressive PCa[29]. We also mated *Kmt9α*[fl/fl] with *Nkx3.1-Cre-ER*[T2(Tg/Tg)]/*Pten*[fl/fl]/*Smad4*[fl/fl] mice to generate *Nkx3.1-Cre-ER*[T2(Tg/Tg)]/*Pten*[fl/fl]/*Smad4*[fl/fl]/*Kmt9α*[fl/fl] mice. By treatment of ten weeks old mice with tamoxifen for eight weeks, we obtained prostate-specific *Pten/Smad4* KO and *Pten/Smad4/Kmt9α* KO mice.

In *Pten/Smad4* KO mice, we observed large prostate tumors (Fig. 4a and Supplementary Fig. 11a) while prostates of Ctrl and *Pten/ Smad4/Kmt9α* KO mice appeared normal (Fig. 4a). Accordingly, prostates of *Pten/Smad4* KO mice weighed more than prostates of Ctrl mice, whereas no significant difference was observed between *Pten/ Smad4/Kmt9α* KO and Ctrl mice (Fig. 4b). Proliferation markers (MKI67 and PCNA) were highly expressed in prostate tumor tissue of *Pten/ Smad4* KO mice but barely detected in prostate tissue of *Pten/Smad4/ Kmt9α* KO and Ctrl mice (Fig. 4c). These findings show that the well-documented prostate tumor formation in *Pten/Smad4* KO mice[29] is severely impaired by concomitant deletion of *Kmt9α*. Using TUNEL assays, we did not detect increased cell death in prostates of *Pten/ Smad4/Kmt9α* KO compared to *Pten/Smad4* KO mice, indicating that *Kmt9α* deletion suppresses tumor growth independently of cell death induction (Supplementary Fig. 11b, c).

To address whether KMT9 regulates DLAT methylation in PCa in mice, we utilized multiplexed immunofluorescence staining to compare distribution and expression of *Kmt9α* as well as levels of DLAT K596me1 in prostate tissue from Ctrl, *Pten/Smad4* KO, and *Pten/Smad4/Kmt9α* KO mice. Our results revealed that mitochondrial KMT9α and methylated DLAT colocalize and that the levels of both proteins are increased in *Pten/Smad4* KO compared to Ctrl prostate tissue (Fig. 4d–f). Importantly, DLAT K596 methylation was almost abolished in *Pten/Smad4/Kmt9α* KO mice. (Fig. 4d, f). Furthermore, mitochondrial KMT9α correlated with mitochondrial DLAT K596me1 levels (Fig. 4g), whereas nuclear KMT9α levels did not (Fig. 4h). This evidence strongly suggests that mitochondrial rather than nuclear KMT9 mediates DLAT K596 monomethylation in vivo. Next, we investigated whether DLAT methylation by mitochondrial KMT9 affected PDC activity and lipogenesis in mice. The comparison of prostate tissue from Ctrl, *Pten/Smad4* KO, and *Pten/Smad4/Kmt9α* KO mice revealed a significant increase in PDC activity, free fatty acid levels, and triglyceride levels in *Pten/Smad4* KO prostates compared to Ctrl tissue (Fig. 4i–k). Of note, the deletion of *Kmt9α* reversed this increase to Ctrl levels. Together, these data show that the occurrence of mitochondrial KMT9 and mitochondrial DLAT K596me1 strongly correlates with KMT9-dependent differences in PDC activity, lipogenesis and prostate tumor growth in vivo.

## Mitochondrial KMT9α and DLAT K596me1 levels are elevated in human PCa samples and correlate with Gleason grade

In the final set of experiments, we investigated by multiplexed immunofluorescence staining whether KMT9α expression and DLAT K596me1 levels were also increased in human PCa samples of various Gleason grades. Our analysis of human prostate tumor samples ($n = 160$) revealed that the levels (H-scores) of mitochondrial KMT9α and DLAT K596me1 were significantly elevated in prostate tissue with higher Gleason grades (grades 4 and 5) compared to normal ($n = 16$) and normal adjacent tissue (NAT, $n = 16$) (Fig. 4l, m). In support of our previous study[17], also nuclear KMT9α levels were increased in higher Gleason-grade tumors (Supplementary Fig. 12a). Furthermore, the staining confirmed the strong co-localization of KMT9 and DLAT K596me1 (Supplementary Fig. 12b), similar to what we observed in mouse tissue. Again, mitochondrial KMT9α and DLAT K596me1 levels correlated (depicted as H-score ratios of DLAT K596me1/DLAT vs. KMT9α/TOMM20) (Fig. 4n), whereas no correlation was observed for nuclear KMT9α and methylated mitochondrial DLAT (Fig. 4o).

In summary, our findings represent an important example of a histone methyltransferase with dual functions in the nucleus and in mitochondria. Our results from cell lines, mice, and human prostate tumor samples consistently show that mitochondrial KMT9 is a key regulator of DLAT methylation controlling DLAT activity, de novo lipogenesis, and PCa growth. This dependency might be therapeutically exploited with mitochondrial KMT9 inhibitors, possible in combination with other treatments such as inhibition of de novo lipogenesis, to selectively target PCa cells.

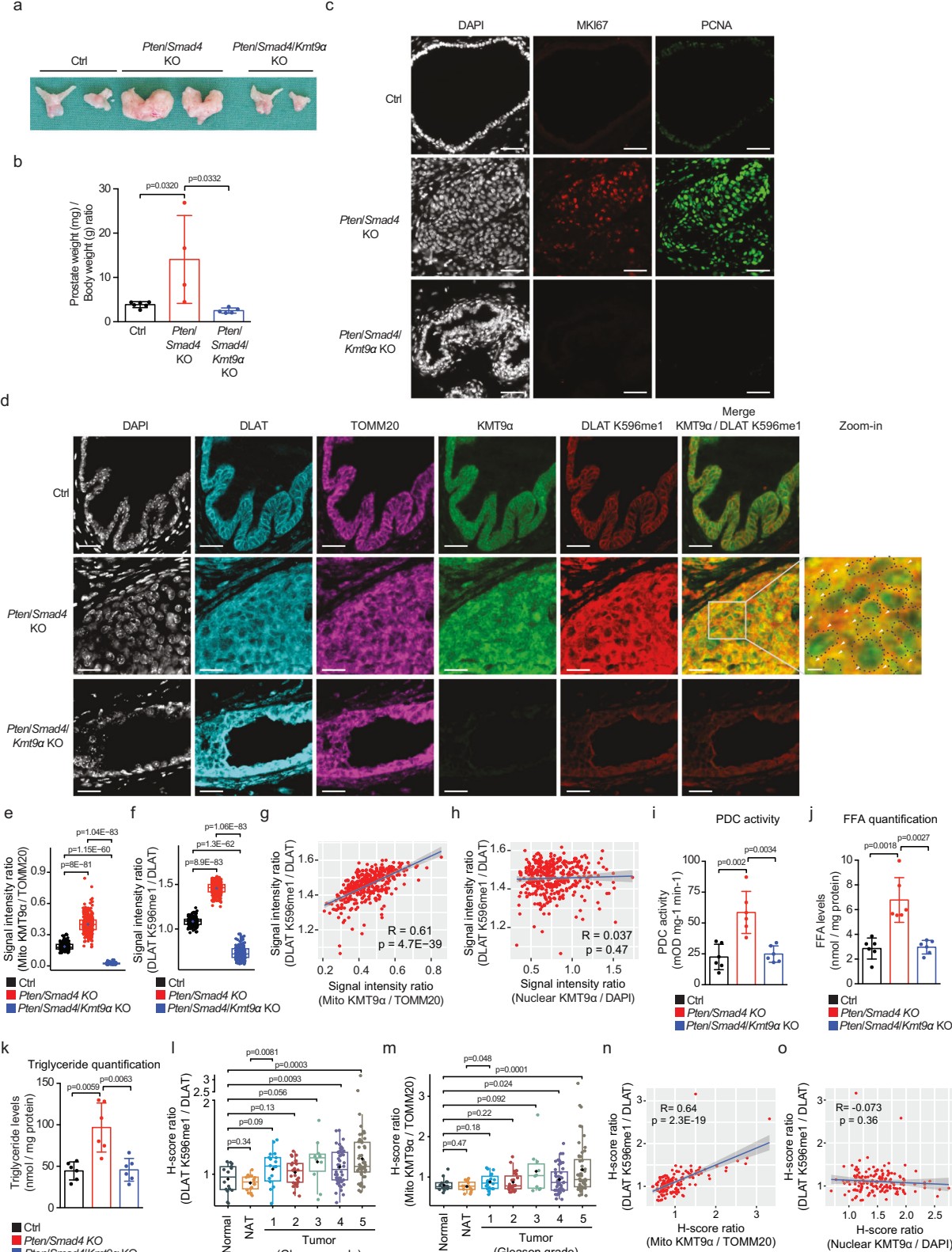

## Discussion

In this study, we uncovered the specific relevance of mitochondrial KMT9 for PDH activity, de novo lipogenesis, and proliferation of PCa cells. KMT9 has previously been shown to regulate cellular processes in the nucleus and the cytoplasm[17,20,22]. An unexpected finding of this study is the apparently exclusive localization of KMT9 in mitochondria of PCa cells and to a lesser extent in normal prostate, but not in

mitochondria of other tumor cell types. The tissue-specific mitochondrial localization of KMT9 in prostate cells aligns with the emerging concept of tissue-specific regulation of mitochondrial functions. Several studies have demonstrated that mitochondrial properties and protein compositions vary significantly across different tissues, reflecting their adaptation to tissue-specific metabolic demands[30]. For instance, the mitochondrial proteome shows substantial tissue-

**Fig. 4 | Mitochondrial KMT9 controls PDC activity and de novo lipogenesis in prostate tumors in mice. a, b** Representative mouse prostates (**a**) and prostate/body weight ratio (mg/g) from Ctrl (*n* = 6 mice), *Pten/Smad4* KO (*n* = 4 mice), and *Pten/Smad4/Kmt9α* KO (*n* = 5 mice) mice (**b**). **c, d** Multiplexed immunofluorescence showing MKI67 and PCNA (**c**) or DLAT, TOMM20, KMT9α and DLAT K596me1 (**d**) in prostate tissue from indicated genotypes. Scale bar: 50 μm. **e, f** Quantification of mitochondrial KMT9α/TOMM20 (**e**) and DLAT K596me1/DLAT (**f**) ratios in prostate sections (Ctrl *n* = 179 cells, *Pten/Smad4* KO *n* = 404 cells, *Pten/Smad4/Kmt9α* KO *n* = 182 cells). **g, h** Pearson correlation between DLAT K596me1/DLAT ratio and mitochondrial KMT9α/TOMM20 ratio (**g**) or nuclear KMT9α/DAPI ratio (**h**) in *Pten/Smad4* KO prostate sections. R, Pearson correlation coefficient. **i–k** Analysis of PDC activity (**i**), free fatty acid levels (**j**), and triglyceride levels (**k**) in prostate tissue of Ctrl, *Pten/Smad4* KO, and *Pten/Smad4/Kmt9α* KO mice (*n* = 6 mice per group). **l, m** H-score ratios of DLAT K596me1/DLAT (**l**) and mitochondrial KMT9α/TOMM20

(**m**) in normal (*n* = 16 tissue microarray cores), normal adjacent tissue (NAT, *n* = 16 tissue microarray cores), and PCa tissues of Gleason grades 1-5 (*n* = 18, 27, 10, 48, 52 tissue microarray cores respectively). **n, o** Pearson correlation between H-score ratios of DLAT K596me1/DLAT and mitochondrial KMT9α/TOMM20 (**n**) or nuclear KMT9α/DAPI (**o**) in prostate tissue sections of human patients. R, Pearson correlation coefficient. Original immunofluorescence images used for the quantification are shown in Supplementary Fig. 12b. **b, e, f, i–m**, Data represent mean ± SD in bar graphs (**b, i–k**) or box plots (**e, f, l, m**; center line: median; box bounds: 25th and 75th percentiles; whiskers: minimum and maximum values within 1.5 times the interquartile range). **b, e–o** Statistical significance was determined using two-sided Student's t-test (**b, e, f, l, i–m**) or Pearson correlation analysis with two-sided p-values calculated based on the t-distribution with n-2 degrees of freedom (**g, h, n, o**). All experiments were independently repeated at least three times with similar results. Source data are provided as a Source Data file.

specific variations, with an estimated 20–30% of mitochondrial proteins being tissue-specific[31,32]. In prostate cells, unique metabolic features have been well-documented, such as altered citrate metabolism and specific lipid synthesis pathways, which distinguish them from other cell types[4,5,33]. These prostate-specific metabolic characteristics might necessitate specialized regulatory mechanisms, including tissue-specific mitochondrial protein localization. However, the exact mechanisms controlling the specific localization of KMT9 in mitochondria of the prostate are currently unclear and require further investigation.

Only a limited number of epigenetic regulators with functions in the nucleus and in mitochondria has been observed[34] including histone acetyltransferases[35–37] and histone deacetylases[38–47]. In this context, KMT9 is an important histone methyltransferase controlling mitochondrial function and thus represents an epigenetic regulator with activities in the nucleus[17–19], the cytoplasm[20–22], and in mitochondria. The majority of proteins in mitochondria are nuclear-encoded, synthesized at ribosomes in the cytoplasm, and subsequently imported into mitochondria[48]. Analyzing the amino acid sequence of KMT9α, we did not detect a putative mitochondrial targeting sequence[48,49], and dissipation of the mitochondrial membrane potential did not block the import into mitochondria, suggesting that mitochondrial import of KMT9α may be achieved by a non-canonical import sequence and/or via alternative mechanisms[35,48,50].

Our results show that mitochondrial KMT9 controls PDC, the gatekeeper of mitochondrial metabolism, by monomethylation of its DLAT subunit at K596. PDC activity is known to be regulated by various post-translational modifications including acetylation, lipoylation, phosphorylation, and O-GlcNAcylation[9–12,15]. KMT9-mediated DLAT monomethylation uncovers an important regulatory modification providing not only an important mechanistic insight into the mode of PDC activity control but also suggests potential for therapeutic opportunities. KMT9 or DLAT depletion in PCa cells compromises de novo lipogenesis and PCa cell proliferation. These observations are in accordance with a recent report by Chen et al.[2] using genetic and pharmacological inactivation of PDHA1 to show the dependency of PCa cells on PDC activity. Importantly, we provide evidence that KMT9-mediated control of PDC activity is not limited to PCa cell lines but is also a relevant mechanism in a well-established PCa mouse model[29] as well as in human patients further underlining the therapeutic potential.

Modulation of PDC activity has attracted considerable attention as a potential treatment option for several diseases including cancer[51,52]. The presence of KMT9 in mitochondria of PCa cells, but not in mitochondria of surrounding non-prostate tissue, presents a potential opportunity to selectively block PDC activity with mitochondrial KMT9 inhibitors. To optimize efficacy, mitochondrial KMT9 inhibitors might be combined with drugs targeting metabolic regulators in other cell compartments and/or inhibitors of nuclear KMT9 function. Together, our findings suggest potential actionable

therapeutic approaches to exploit the dependency of PCa cells on mitochondrial metabolism.

## Methods

### Study approvals
All animal experiments were conducted in compliance with relevant ethical regulations and guidelines. The study protocol was approved by the Regional Board of Freiburg (Regierungspräsidium Freiburg, approval number: 35-9185.81/G-23/011). According to institutional guidelines, mice were euthanized when weight loss reached 20% of initial body weight or when they showed signs of severe health deterioration.

### Mouse studies
All mice were housed in the pathogen-free barrier facility of the University Medical Center Freiburg. Mice were maintained under temperature- and humidity-controlled conditions with a 12-h light/dark cycle, free access to water, and a standard (3807, Granovit, Kaiseraugst, Switzerland) or 400 mg/kg tamoxifen citrate-containing (A115T70400, Ssniff, Soest, Germany) rodent chow. Nkx3.1-Cre-ER$^{T2(Tg/Tg)}$/*Pten*$^{fl/fl}$/*Smad4*$^{fl/fl}$, Nkx3.1-Cre-ER$^{T2(Tg/Tg)}$/*Pten*$^{fl/fl}$/*Smad4*$^{fl/fl}$/*Kmt9a*$^{fl/fl}$ mice (10 weeks) were used for in vivo experiments. After 8 weeks of tamoxifen diet treatment, these mice generated *Pten/Smad4* KO and *Pten/Smad4/Kmt9α* KO mice, respectively. Control mice were Nkx3.1-Cre-ER$^{T2(Tg/Tg)}$/*Pten*$^{fl/fl}$/Smad4$^{fl/fl}$ mice without tamoxifen diet treatment (*n* = 6 for Ctrl group, *n* = 4 for Pten/Smad4 KO and *n* = 5 for Pten/Smad4/Kmt9α KO). Only male mice were used since prostate is a male-specific organ. Animals were killed by cervical dislocation and tissues immediately processed for further analyses.

### Cell lines and cell culture
All cell lines used in this study were obtained from commercial sources: DU145 (American Type Culture Collection, cat. no.: HTB-81), PC-3M (Tebu-Bio, cat. no.: 305061), LNCaP (German Collection of Microorganisms and Cell Cultures, cat. no.: ACC 256), C4-2B (Thermo Fisher Scientific, cat. no.: 50-238-4833), HepG2 (American Type Culture Collection, cat. no.: HB-8065), SW480 (European Collection of Cell Cultures, cat. no.: 87092801), HT1376 (Tebu-Bio, cat. no.: 305100), MCF10A (Merck, cat. no.: CLLS1069), and Panc-1 (Thermo Fisher Scientific, cat. no.: 50-238-2594). DU145, PC-3M, LNCaP, and C4-2B cells were maintained in RPMI 1640 medium. SW480, HT1376, and HepG2 cells were cultured in DMEM. MCF10A cells were cultured in DMEM/F12 medium supplemented with 20 ng/ml EGF, 0.5 μg/ml hydrocortisone, 100 ng/ml cholera toxin, and 10 μg/ml insulin. Panc-1 cells were maintained in DMEM high glucose medium. All media were supplemented with 10% fetal calf serum and penicillin/streptomycin. All cell lines were regularly tested and confirmed to be free of mycoplasma contamination.

## Plasmids, siRNAs, and transient transfection

All plasmids and siRNAs used in this study are listed in Supplementary Data 3. Detailed information on cloning procedures will be provided upon request. Plasmids for expression of KMT9α or mutant KMT9α (N122A) containing an N-terminal MTS (MLFNLRILLN-NAAFRNGHNFMVRNFRCGQPLQ) or NLS (PKKKRKV) sequence were generated by PCR cloning. Cells were transfected with siCtrl, siKMT9α or siKMT9β using DharmaFECT™ 1 Transfection Reagent (Thermo Scientific) according to the manufacturer's instructions. For rescue experiments, cells were simultaneously transfected with expression plasmids and siRNAs using Lipofectamine™ 3000 Transfection Reagent (Thermo Scientific).

## Isolation of primary mouse prostate epithelial cells

Mouse prostate tissues were mechanically minced and enzymatically dissociated using a tissue dissociation kit (Miltenyi Biotec) in DMEM. The tissue suspension was processed using a MACS Octo Dissociator (Miltenyi Biotec) with temperature control, followed by manual disaggregation through a 20 G needle. The resulting cell suspension was sequentially filtered through 70 µm and 30 µm cell strainers. After centrifugation at $300 \times g$ for 7 minutes, cells were resuspended in DMEM and viability was assessed using Trypan blue staining. Mouse primary prostate epithelial cells were isolated by FACS based on CD49f-low and EpCAM-high expression. The cells were cultured in Advanced DMEM/F-12 medium supplemented with penicillin/streptomycin, 10 mM HEPES, 2 mM Glutamine, B-27 (1×), N-acetylcysteine (1.25 mM), A83-01 (0.2 µM), DHT (1 nM), Noggin (0.1 µg/ml), mouse R-spondin-1 (0.5 µg/ml), mouse EGF (50 ng/µl), and Y-27632 (10 µM).

## Recombinant proteins

Expression and purification of recombinant His-KMT9α/KMT9β, His-KMT9α (N122A)/KMT9β, and NLS-KMT9α-His/KMT9β were performed as previously described[17]. Briefly, protein expression in E. coli BL21-CodonPlus-RIPL was induced with 1 mM IPTG overnight at 16 °C. Bacteria were disrupted in buffer 1 [25 mM Tris-HCl (pH 8.0), 200 mM NaCl, 5% glycerol, pH 8.0, 5 mM beta-mercaptoethanol, cOmplete™ EDTA-free Protease Inhibitor Cocktail (Roche)] supplemented with 10 mM imidazole with an EmulsiFlex high pressure homogeniser (Avestin). Proteins were affinity-purified using Ni-NTA fast flow resin (Qiagen). The resin was washed with buffer 1 supplemented with 20 mM imidazole, and proteins were eluted with buffer 2 [25 mM Tris-HCl (pH 8.0), 200 mM NaCl, 5% glycerol, 250 mM imidazole (pH 8.0), 5 mM beta-mercaptoethanol]. Eluted fractions were dialyzed against assay buffer [10 mM Tris-HCl (pH 8.0), 100 mM NaCl, 20% glycerol, 2 mM DTT], aliquoted, flash frozen in liquid nitrogen, and stored at -80 °C. Recombinant human DLAT protein was purchased from Sino Biological Inc.

## Immunofluorescent staining

Cells were seeded on coverslips in 24-well plates and stained by adding 200 nM MitoTracker™ Red CMXRos (ThermoFischer) at 37 °C for 30 min. Stained cells were fixed with 4% formaldehyde for 10 min, permeabilized with 0.5% Triton X-100 in PBS followed by treatment with blocking buffer [PBS, 1% bovine serum albumin, 0.05% Triton X-100, 0.2% Tween® 20, 5% FBS] for 1 hour at room temperature. Fixed cells were incubated with primary antibody diluted in blocking buffer overnight at 4 °C, washed, and incubated with secondary antibody for 1 hour. Nuclei were stained with DAPI. Finally, coverslips were washed and mounted in Fluoromount-G (Southern Biotech). Fluorescence images were acquired using a LSM 880 laser scanning microscope (Carl Zeiss Micro Imaging) and analyzed with ImageJ (National Institutes of Health) or Zeiss ZEN Microscope Software (Carl Zeiss Micro Imaging).

## Cell fractionation and isolation of mitochondria

Cell fractionation and isolation of mitochondria were performed using the Mitochondria Isolation Kit for Cultured Cells (Abcam) according to the manual's instructions. Briefly, cells were frozen and thawed once in liquid nitrogen, resuspended in Reagent A, incubated for 10 min on ice, and homogenized with a Dounce homogenizer. The homogenate was centrifuged at $1000 \times g$ for 10 min at 4 °C and the supernatant preserved (supernatant #1). The pellet was resuspended in Reagent B and homogenized. The homogenate was centrifuged at $1000 \times g$ for 10 min at 4 °C and the supernatant preserved (supernatant #2). The pellet containing nuclei was lysed in RIPA buffer [150 mM NaCl, 50 mM Tris-HCl (pH 7.4), 1% NP40, 0.5% sodium deoxycholate and 0.1% SDS] (nuclear fraction). Supernatants #1 and #2 were combined, mixed thoroughly, and centrifuged at $12,000 \times g$ for 15 min at 4 °C. The supernatant was preserved (cytosolic fraction). The pellet was lysed with RIPA lysis buffer (mitochondrial fraction).

## Proteinase K protection assay

Mitochondria were isolated as described above except that mitochondrial pellets were resuspended in each of the three following buffers: (1) isotonic buffer [10 mM MOPS-KOH, (pH 7.2), 250 mM sucrose, 1 mM EDTA], (2) hypotonic buffer [10 mM MOPS-KOH (pH 7.2), 1 mM EDTA] or (3) 1% Triton buffer [10 mM Tris-HCl (pH 7.4), 1 mM EDTA, 1% Triton X-100] followed by treatment with 50 µg/ml proteinase K for 15 min on ice. The reaction was terminated by addition of 20 mM PMSF. Next, proteins were precipitated with 10% trichloroacetic acid (TCA), pellets washed with acetone, resuspended in 1× SDS loading buffer, and denatured at 95 °C for 10 min. Finally, the samples were subjected to Western blot analysis.

## Mitochondrial import assay

Mitochondria were isolated by differential centrifugation as described previously[53] with the following modifications. Cells were washed twice with PBS and resuspended in homogenization buffer A [83 mM sucrose, 10 mM HEPES (pH 7.2)]. After cell disruption with a glass-teflon homogenizer, an equal volume of buffer B [250 mM sucrose, 10 mM HEPES (pH 7.2)] was added. Samples were centrifuged for 5 min at $1000 \times g$ to remove the nuclear pellet. Mitochondria present in the supernatant were pelleted by centrifugation for 5 min at $12,000 \times g$ and pellets were washed with buffer C [320 mM sucrose, 1 mM EDTA, 10 mM Tris-HCl (pH 7.4), 1 mM PMSF]. Radiolabelled KMT9α and KMT9β were produced by in vitro transcription with the mMessage mMachine T7 kit (Ambion) using pCMX-DEST51-KMT9α-6xHis and pETDuet1-KMT9β-6xHis as templates followed by translation using TNT coupled reticulocyte lysate (Promega) in the presence of [$^{35}$S]-methionine. Prior to import, the lysates containing radiolabelled proteins were incubated at 30 °C for 10 min in the presence of 5 mM ATP and 1 mM DTT and cleared by centrifuging at $12,000 \times g$ for 10 min. Import into freshly isolated mitochondria was performed at 30 °C in import buffer [3% (w/v) bovine serum albumin, 250 mM sucrose, 5 mM magnesium acetate, 80 mM potassium acetate, 10 mM sodium succinate, 1 mM DTT, 5 mM ATP, 20 mM HEPES-KOH (pH 7.4)][54]. For -Δψ control samples, the membrane potential was dissipated by addition of 1 µM valinomycin. Import reactions were stopped by addition of 1 µM valinomycin and transferred to ice. Where indicated, non-imported protein was degraded by incubation with 50 µg/ml proteinase K for 15 min on ice. Mitochondria were re-isolated and washed using import buffer without BSA and import was assessed by SDS-PAGE analysis and autoradiography.

## Immunoprecipitation and Western blotting

For immunoprecipitation (IP) of KMT9α or DLAT, cells or freshly isolated mitochondria were lysed in IP buffer [150 mM NaCl, 20 mM Tris-HCl (pH 8.0), 0.15% NP40, cOmplete™ Protease Inhibitor Cocktail] for

10 min followed by sonication for 90 seconds using a Bandelin Sonorex RK 52 H. Samples were centrifuged and supernatants incubated with antibody overnight at 4 °C followed by 2 hours of incubation with Dynabeads™ Protein G (Invitrogen). Beads were washed five times with IP buffer, dissolved in 1× SDS-PAGE sample buffer, and subjected to Western blot analysis as previously described[17]. For IP of PDC, freshly isolated mitochondria were lysed in lysis buffer [PBS containing 1% lauryl maltoside and cOmplete™ Protease Inhibitor Cocktail] for 30 min on ice. Extracts were cleared by centrifugation at $16,000 \times g$ for 10 min at 4 °C and supernatants incubated with PDH Immunocapture Kit beads (Abcam) for 3 hours at room temperature. Beads were washed three times with buffer (PBS containing 0.05% lauryl maltoside), and PDC was eluted with reducing reagent-free Laemmli buffer. All antibodies used for IP and Western blotting are listed in Supplementary Data 3. All experiments were done in triplicates with representative images shown in the figures.

### Characterization of the KMT9 interactome

Firstly, total cell lysate of PC-3M cells was subjected to KMT9α immunoprecipitation with anti-KMT9α or IgG as negative control, following the protocol described above in the section of 'Immunoprecipitation and Western blotting'. After the IP, beads were washed three times with 50 mM $NH_4HCO_3$ and incubated with 10 ng/µl trypsin in 1 M urea 50 mM $NH_4HCO_3$ for 30 minutes, washed with 50 mM $NH_4HCO_3$ and the supernatant digested overnight (ON) in presence of 1 mM DTT. Digested peptides were alkylated and desalted prior to LC-MS analysis. For LC-MS/MS purposes, desalted peptides were injected in an Ultimate 3000 RSLCnano system (Thermo), separated in a 15-cm analytical column (75µm ID with ReproSil-Pur C18-AQ 2.4 µm from Dr. Maisch) with a 50-min gradient from 4 to 40% acetonitrile in 0.1% formic acid. The effluent from the HPLC was directly electrosprayed into a Qexactive HF (Thermo) operated in data-dependent mode to automatically switch between full scan MS and MS/MS acquisition. Survey full scan MS spectra (from m/z 375–1600) were acquired with resolution R = 60,000 at m/z 400 (AGC target of $3\times10^6$). The 10 most intense peptide ions with charge states between 2 and 5 were sequentially isolated to a target value of $1\times10^5$, and fragmented at 27% normalized collision energy. Typical mass spectrometric conditions were: spray voltage, 1.5 kV; no sheath and auxiliary gas flow; heated capillary temperature, 250°C; ion selection threshold, 33.000 counts. MaxQuant (version 1.6.6.0 53) was used to identify proteins and quantify by iBAQ with the following parameters: Database, UP000005640_Hsapiens_190109; MS tol, 10ppm; MS/MS tol, 20ppm Da; Peptide FDR, 0.1; Protein FDR, 0.01 Min. peptide Length, 7; Variable modifications, Oxidation (M); Fixed modifications, Carbamidomethyl (C); Peptides for protein quantitation, razor and unique; Min. peptides, 1; Min. ratio count, 2. Statistical significance of the differences (anti-KMT9α vs IgG) in protein abundance was calculated with Perseus software Perseus (version 2.1.3.0) using FDR adjustment of a Student's t test. Identified proteins were considered as interaction partners of the bait if the FDR value is less than 0.05.

### Metabolomic analysis

Global untargeted metabolomics analysis was performed with siCtrl and siKMT9α-treated PC-3M cells ($n$ = 6 biological replicates per condition). Samples were prepared using automated MicroLab STAR® system (Hamilton). Proteins were precipitated with methanol under vigorous shaking followed by centrifugation. The resulting extract was divided into five fractions for different analytical methods. All analyses were performed using a Waters ACQUITY UPLC system coupled to a Thermo Scientific Q-Exactive high resolution/accurate mass spectrometer with a heated electrospray ionization (HESI-II) source and an Orbitrap mass analyzer operated at 35,000 mass resolution. Separations were performed on C18 columns

(Waters UPLC BEH C18-2.1×100 mm, 1.7 µm) and HILIC columns (Waters UPLC BEH Amide 2.1×150 mm, 1.7 µm) with various mobile phase compositions. For C18 separations, three different conditions were used: (1) water and methanol containing 0.05% PFPA and 0.1% FA for hydrophilic compounds under acidic positive mode, (2) methanol, acetonitrile, water with 0.05% PFPA and 0.01% FA for hydrophobic compounds under acidic positive mode, and (3) methanol and water with 6.5 mM ammonium bicarbonate at pH 8 for basic negative mode. For HILIC separation, water and acetonitrile with 10 mM ammonium formate at pH 10.8 were used. Data were acquired using a Thermo Scientific Q-Exactive high resolution/accurate mass spectrometer with a HESI-II source and an Orbitrap mass analyzer operated at 35,000 mass resolution. The MS analysis alternated between MS and data-dependent $MS^n$ scans using dynamic exclusion, covering a scan range of 70-1000 m/z. Raw data files were processed using Metabolon's proprietary software (version 7) for peak detection and integration. Compounds were identified by comparison to library entries containing retention time, mass-to-charge ratio, and MS/MS spectral data. Library matching was based on a retention index within a narrow RI window, accurate mass match within 10 ppm, and MS/MS forward and reverse scores. Quality control (QC) samples were prepared by pooling equal aliquots from all experimental samples to create a representative pooled QC sample. Technical replicates derived from the pooled QC sample were analyzed multiple times over the course of the analysis, with multiple replicates included within each experimental batch. Metabolon uses pooled QC sample data to evaluate platform performance and calculate the median relative standard deviation (RSD) across a study. Data were normalized to Bradford protein concentration, log transformed and imputation of missing values, if any, with the minimum observed value for each compound. Statistical analysis was performed using ArrayStudio (version 10.0) and R (version 4.2.0) on log-transformed data. Welch's two-sample t-test was used to identify biochemicals that differed significantly between experimental groups ($p \leq 0.05$).

### Targeted detection of DLAT methylation by mass spectrometry

Targeted detection of DLAT lysine methylation was performed using the Complete360® multi-omics pipeline (Complete Omics Inc.) with minor modifications. Briefly, immunoprecipitated PDC proteins were eluted from beads, denatured, alkylated, and digested as previously described[17,55]. Peptidome samples were cleaned using an in-house packed C18 micro-elute 96-well plate with 0.1% TFA as the washing solution and 40% acetonitrile 0.1% TFA as the elution solution. Optimal detection parameters for methylated and unmethylated peptides were established through a massive parallel optimization procedure, in which all potential detectable precursor and fragmented ions were optimized for their collision energy values with minimal step (1 eV) sized ramping and optimized for their source parameters with a single Dalton mass tolerance. Optimal detection parameters were collected as previously described[55]. Peptidome samples were analyzed on an Agilent 6495 Triple Quadrupole LC/MS mass spectrometer with in-house hardware modifications boosting ionization efficiency. Data analysis was performed with Skyline (version 24.1)[56].

### PDC activity assay

PDC activity assays were performed using the Pyruvate Dehydrogenase (PDH) Enzyme Activity Microplate Assay Kit (Abcam) according to manufacturer's instructions. Briefly, fresh mitochondrial extracts were dispensed into 96-well assay plates pre-coated with anti-PDC monoclonal antibody. After 3 hours of incubation at room temperature, the assay plates were subjected to sequential removal of liquid extract, washing with 1× stabilizer, and addition of freshly prepared reaction buffer (mixture of 20× reagent mix, 1× buffer, coupler,

and reagent dye). PDC enzymatic activity was measured at 450 nm using a TECAN Infinite M200 plate reader.

## Cell proliferation assay

Cell proliferation was determined using the X-Celligence RTCA system (Roche). For real-time recording of cell proliferation, DU145 (6000 cells/well), PC-3M (3000 cells/well), LNCaP (20,000 cells/well), HepG2 (10,000 cells/well), SW480 (5000 cells/well), HT1376 (5000 cells/well), MCF10A (5000 cells/well), and Panc-1 (10,000 cells/well) cells were seeded into 16-well E-plates (Roche). Cells were transfected with siRNA at a final concentration of 80 nM using DharmaFECT™ 1 Transfection Reagent (Thermo Scientific) for 24 hours prior to seeding. For rescue experiments, cells were co-transfected with 80 nM siRNA and 1.25 ng/μl expression plasmid using Lipofectamine™ 3000 Transfection Reagent for 24 hours prior to seeding. For co-treatment of siKMT9α and lipogenesis inhibitor, cells were transfected with siKMT9α at a low concentration (2.4 to 10 nM for DU145, PC-3M, LNCaP, HepG2, SW480, HT1376, MCF10A, or Panc-1 cells, respectively) for 24 hours and cell proliferation was recorded in the presence or absence of 100 μM SB204990 or 60 μM NDI-091143. Cell indices were automatically recorded every 15 min. Relative velocities represent the change of the cell index over time.

## Human tissue acquisition

Healthy colon tissue as well as mesenchymal tissue specimens from male donors were procured during laparoscopic colon surgery. The identification and classification of non-neoplastic tissue were conducted through macroscopic evaluation in situ. Tissue samples derived from subjects with a prior diagnosis of colorectal adeno-carcinoma; however, the specimens were specifically excised from regions proximal to the neoplastic lesion and were determined to be free of malignant tumor mass. Mesenchymal tissue was obtained from mesocolonic tissue in healthy regions. Procedures were executed in adherence to local guidelines of the Ethics Committee of the University of Freiburg (ETK: 21-1162_5). The bladder (obtained from a male donor) and prostate cancer specimens were acquired by transurethral resection. For bladder cancer samples the identification of cancerous tissue was based on a macroscopic assessment. For prostate cancer samples the tissue condition was predefined by a previous cancer diagnosis. All donors provided informed consent prior to the procurement of tissue samples. Procedures were executed in adherence to local guidelines of the Ethics Committee of the University of Freiburg (ETK: 266/14).

## Multiplexed immunofluorescence analysis

Prostate tissues from Ctrl, prostate-specific *Pten/Smad4* KO, and *Pten/Smad4/Kmt9α* KO mouse models were collected and fixed in 4% paraformaldehyde. Sections of 5 μm thickness were obtained using a microtome and mounted onto glass slides. Deparaffinization was achieved by heating slides on a heating plate at 55-60 °C until paraffin melted, followed by sequential incubation in xylene and ethanol gradient. Multiplex immunofluorescence staining was performed using the Opal 7-color kit (Akoya Biosciences) according to the manufacturer's instructions with slight modifications. Briefly, antigen retrieval was achieved by heating sections in AR9 Buffer in a pressure cooker for 20 minutes. Next, slides were blocked with Antibody Diluent, then incubated with primary antibodies against the first marker (KMT9α) overnight at 4 °C, followed by secondary antibody incubation, Opal fluorophore incubation, and antibody removal. Subsequent markers (DLAT K596me1, DLAT, TOMM20, respectively) were stained using the same protocol. Spectral DAPI staining was conducted after completing all marker stainings. Slides were mounted in Fluoromount and imaged using a PhenoImager™ Fusion (Akoya Biosciences) for image analysis of protein expression and distribution in prostate tissue samples.

## TUNEL assay

TUNEL assay was performed to detect cell death using the In Situ Cell Death Detection Kit (Cat. #11684795910, Roche). Following deparaffinization and rehydration, sections were treated with proteinase K (20 μg/ml in 10 mM Tris/HCl, pH 7.4-8.0) for 30 min at room temperature. After rinsing twice with PBS, positive control sections were pre-treated with DNase I (50 U/ml) in 1× DNase buffer (Promega) for 10 min at room temperature. TUNEL reaction mixture was prepared by combining 50 μl enzyme solution with 450 μl label solution. Sections were incubated with TUNEL reaction mixture for 60 min at 37 °C in a humidified atmosphere in the dark. Following three PBS washes, nuclei were counterstained with DAPI (0.5 μg/ml in PBS) for 10 min at room temperature. Slides were washed twice with PBS and mounted using Fluoromount mounting medium. TUNEL-positive cells and nuclei were visualized in green and blue channels, respectively.

## Tissue microarrays analysis

Human prostate tissue microarrays containing normal prostate, normal adjacent, and PCa tissue were purchased from TissueArray.Com LLC (cat. No.: PR1921c) and utilized for multiplex immunofluorescence staining to assess the expression levels and distribution patterns of KMT9α, DLAT K596me1, DLAT, and TOMM20. Staining was performed using the Opal 7-color kit (Akoya Biosciences) following the same protocol as described earlier for multiplex immunofluorescence staining in mouse tissue. Image analysis was performed using QuPath software (version 0.5.0). The H-score[57], was utilized to evaluate staining intensity of each marker. For H-score calculations, staining intensities were assigned to four levels: negative, weak, moderate, and strong corresponding to scores of 0, 1, 2, and 3, respectively. Then the percentage of cells at each staining intensity level relative to the total number of cells was determined. The H-score was calculated by multiplying the percentage of cells at each staining intensity level with the corresponding score. The sum of these values represents a comprehensive assessment of protein expression levels on a scale of 0 to 300.

## Statistical analysis

Results from quantitative analyses are presented as mean ± standard deviation (SD), mean + SD or as box plots showing median (center line), 25th and 75th percentiles (box bounds), and minimum and maximum values within 1.5 times the interquartile range (whiskers). Statistical analyses were performed using the Student's t-test with two-tailed distribution in Excel (Microsoft 365) and GraphPad Prism (version 7.00). Sample sizes are specified in figure legends.

## Reporting summary

Further information on research design is available in the Nature Portfolio Reporting Summary linked to this article.

## Data availability

The targeted mass spectrometry data for DLAT methylation detection presented in supplementary Fig. 3a have been deposited in the PeptideAtlas SRM Experiment Library (PASSEL) via ProteomeXchange under accession code PASS05878. The LC-MS/MS data for KMT9α interactome analysis presented in Fig. 1f have been deposited to the ProteomeXchange Consortium via the PRIDE partner repository under accession code PXD053942. The metabolomics data in Fig. 1h−l have been deposited to the MetaboLights database with the dataset identifier MTBLS2452. The remaining data are available within the Article, Supplementary Information or Source Data file. Source data are provided with this paper.

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

## Acknowledgements

We thank Astrid Rieder, Helena Fischer, Lioba Walz for technical assistance. We thank the Lighthouse Core Facility staff of the Medical Center University of Freiburg for helping with multiplexed immunofluorescence staining and their excellent support. This work was supported by grants of the Deutsche Forschungsgemeinschaft - Project ID 192904750 - SFB 992 Medical Epigenetics, Schu688/15-1, and of the European Research Council (ERC AdGrant 322844) to R.S. N.P. was supported by Deutsche Forschungsgemeinschaft grant PF 202/9-1 - Project ID 394024777. U.S. was supported by Deutsche Forschungsgemeinschaft - SFB 1381- project ID 403222702. E.M. was supported by Deutsches Konsortium für Translationale Krebsforschung grant DKTK FR01-374. The Lighthouse Core Facility is funded in part by the Medical Faculty, University of Freiburg (Project Numbers 2023/A2-Fol; 2021/B3-Fol), the DKTK, and the DFG (Project Number 450392965).

## Author contributions

R.S. and Y.J. generated the original hypothesis and designed the experiments. Y.J., J.M.M., M.S., S.U., H.B., H.R., A.I., I.F., S.W., and E.M. performed experiments. Q.W. and U.S. performed the mass spectrometry analysis. K.M.S., A.S., C.B., and C.G. provided the study materials. Y.J., E.M., H.G., and R.S. analyzed the data and wrote the manuscript. R.S., Y.J., E.M., H.G., J.M.M., and N.P. provided intellectual contributions throughout the project. All authors discussed the results and edited the manuscript.

## Funding

## Competing interests

The authors declare no competing interests.
