## [Transparent Peer Review file · Nature Communications]

Mitochondrial KMT9 methylates DLAT to control pyruvate dehydrogenase activity and prostate cancer growth

Corresponding Author: Professor Roland Schüle

Version 1:

Reviewer comments:

Reviewer #1

(Remarks to the Author)

The manuscript by Jia et al reported a specific role of mitochondrial KMT9 in prostate cancer cells for PDH activity, de novo lipogenesis, and cell proliferation. Mechanistically, they showed that this is achieved through mitochondrial KMT9 monomethylating DLAT at lysine 596. Evidence is provided using prostate cancer cell lines, a genetic mouse model, and human prostate specimens. This is an important study that reveals how a histone methyltransferase could regulate mitochondrial metabolism, which in turns influences prostate cancer progression. If these effects are truly specific to prostate cancer cells but not other tissues, it may represent a good therapeutic opportunity against prostate cancer by developing mitochondrial KMT9 inhibitors.

Suggestions for revision:

Fig. 1: Is KMT9 present in mitochondria of primary prostate tissues? For example, the authors could test this using primary wildtype mouse prostate cells, and compare with expressions in primary tissues of other organs.

Fig. 2e: No description of the properties of the human healthy mesenchyme or colon tissues or bladder and prostate cancer specimens were provided. How the tissues were acquired is also not mentioned.

Fig. 3: Experiments were done mostly using prostate cancer cell lines. To show the effects are specific to prostate cancer, control cell lines from other cancer types would be desired. While HepG2 cells were used for Fig. 3n, other figure panels were prostate-only. For example, what will the effects be if MTS-KMT9 α is overexpressed in other cancer cell lines?

Fig. 3: To provide further evidence about cell proliferation, some xenografting experiments would be nice to add to the in vitro assays.

Extended Fig. 3: While previous figures were done using the PC-3 cell line alongside DU145, this figure used LNCaP. Why not PC-3?

Fig. 4c: It is hard to believe that control prostate showed no Ki67 staining. Even for normal prostate homeostasis, a minority of epithelial cells should be Ki67-positive.

Fig. 4: Does KMT9 conditional KO affect cell death rate in the tumor model?

Reviewer #2

(Remarks to the Author)

In the authors' published paper (Metzger, Wang et al., Nat Struct Mol Biol 2019), they identified the 7- β -strand protein KMT9 as a novel histone H4K12 mono-methyltransferase and reported that KMT9 controls prostate cancer (PCa) cell proliferation in an enzymatic activity-dependent manner. There is a lack of correlation between H4K12me1 global levels and cellular phenotypes, implying other targets are involved in KMT9's function in PCa. In the current manuscript, the authors found KMT9 is localized in mitochondria specifically in PCa cells but not in other cancer cell types examined. Using carefully

designed and extensive experiments, they demonstrated that mitochondrial KMT9 mono-methylates lysine 596 of the DLAT subunit in the pyruvate dehydrogenase complex (PDC). Further K596me1 on DLAT controls PDC activity, de novo lipogenesis and PCa cell proliferation. Finally, the authors showed that KMT9 plays an essential role in PDC activity, lipogenesis and tumor growth using a PCa mouse model. Overall, this manuscript is well prepared, and data support their conclusions. We have only a few comments.

1. In Fig. 1 and Fig. 2, most experiments and conclusive data were derived from the PC-3M cell line. However, this cell line is not included in Fig. 3. Does the PC-3M cell line show any phenotype when DLAT K596me1 is disrupted?
2. The conclusion related to Fig. 4 “mitochondrial KMT9 monomethylates DLAT at K596 to regulate PDC activity and lipogenesis in PCa in vivo” is overstated, because data in Fig. 4i-k are insufficient to support the role of DLAT K596me1 in PDC activity and lipogenesis in vivo.
3. Fig. 4m-l: the y-axis can be stretched to better display differences between groups.
4. In the authors' previous paper, they showed that the percentage of PCa samples with nuclear KMT9 staining increased along with disease progression. Do authors also see elevated nuclear KMT9 levels in prostate tissue with higher Gleason grades?
5. Please discuss why mitochondrial localization of KMT9 is only found in PCa.
6. Both KMT9 and PDC are found in nuclei, where DLAT K596me1 cannot be detected. Can authors comment on this?

Kai Ge (NIH) and Guojia Xie (NIH)

Reviewer #3

(Remarks to the Author)

Reviewer #4

(Remarks to the Author)

Review Jia et al. Nature Communications 2024

In this Ms, Jia and collaborators describe a previously unknown effect of the methyl transferase KMT9 on the control of the pyruvate dehydrogenase complex (PDC). The authors made several original observations and identified DLAT (the E2 subunit of the PDC) as the target of mitochondrial KMT9. I believe this study provides interesting molecular insights on the regulation of the PDC, a key and central metabolic hub functioning at the crossroad of glycolysis and the TCA cycle. The data are globally very convincing and well presented, although several points need to be clarified.

1- Could the authors try to estimate the size of the mitochondrial pool of KMT9 based on their immunoblots? Although their biochemical cell fractionation, in organello and fluorescence microscopy assays are consistent with KMT9 / mitochondrial localization, I think that electron microscopy data could further reinforce their conclusion.

2- Did the authors try to identify the domain involved in KMT9 mitochondrial import?

3- Fig 1f. The data related to the identification of KMT9 interactors are very limited both in term of experimental design and description of the results. A table indicating the different interactors identified by LC-MSMS, integrating at least the number and sequence of the identified peptides, and the corresponding statistical analyses must be included. Furthermore, I think it would have been more logical to show the co-IP data confirming the KMT9-DLAT interaction (shown in Fig. 2b) in this panel.

4- One important question relates to the relevance of this mechanism of regulation of the PDC in the nucleus. This question is addressed in their Ms but I believe a few controls are missing. KMT9 was initially described as a nuclear chromatin remodeller. Since a nuclear pool of the PDC has been described by at least 2 independent groups (including one which highlighted its importance in histone acetylation in prostate cancer cells (Chen et al., Nat Genetics 2018)), I think that it is essential that the authors investigate in more details a potential contribution of this KMT9-PDC crosstalk in the nucleus. Although the authors have performed a complementation assay with a NLS-fused version of KMT9, I don't think that this is sufficient to prove that there is no impact of KMT9 on nuclear PDC. Indeed, despite the authors show that this NLS-fused ectopic KMT9 is needed to fully rescue the proliferation defect of KMT9-depleted cells, they don't demonstrate that this NLS-KMT9 can still target DLAT and thereby regulate its potential effects on histone acetylation in prostate cancer cells. The authors should demonstrate that the NLS-peptide is not interfering neither with KMT9 methylation activity on DLAT (using their in vitro assay) nor with DLAT interaction in these cells (by co-IP experiments). The absence of an effect of this NLS-KMT9 on nuclear PDC could also be confirmed by immunoblotting of histone acetylation, in particular on histone H3K9Ac

that was linked to PDC activity in prostate cancer cells by Chen and collaborators. Complementary to data shown in Fig 2f, other experimental evidence that would rule out a contribution of this nuclear DLAT-KMT9 crosstalk could be provided by immunofluorescence experiments using their anti-K596me1 antibody to show that the staining is restricted to mitochondria and is completely excluded from the nucleus in prostate cancer cells. A partial answer to this question is provided by their nice in situ data shown in Fig. 4d where we see poor nuclear staining of DLAT K596me1 but I think that confocal microscopy analyses at high magnification in prostate cancer cells would provide a more definitive answer to this question.

5- I don't really understand why inhibition of KMT9 and ACLY would synergize if they act in the same pathway. The ACLY inhibitor and dKMT9 depletion should recapitulate the same effects except if they used doses that do not inhibit completely their respective target. If not too difficult in their environment, can they provide quantitative data such as stable isotope tracing data using ¹³C-labelled pyruvate) and comment on that?

Is the partial rescue of proliferation provided by MTS-KMT9 (but not by the NLS-KMT9) in KMT9-depleted cells abrogated by the ACLY inhibitor? Is the proliferation defect of KMT9 cells rescued (at least partly) by addition of oleate?

Minor comments:

- Fig 1e: the authors should show LMNA immunoblots to demonstrate the purity of their mitochondrial fraction and the absence of contamination with nuclear material.

- Fig 2a: The authors should not indicate in the legend of this panel that they are detecting DLAT methylation per se with this IP-Western assay. Although this is likely the case, it is impossible to formally prove that the anti-pan methyl antibody recognizes DLAT but not another subunit of the immunoprecipitated PDC in this assay. The DLAT immunoblot only shows that DLAT is indeed present in their immunoprecipitated PDC complex. Only their MS assay and the experiments based on the K596 mutant demonstrate DLAT monomethylation by KMT9.

- Extended Fig 2j. I think the authors should change their conclusion regarding potential crosstalks between KMT9 and other known regulatory mechanisms impinging on the PDC. Indeed, although they conclude that they found "no significant changes of the expression levels of PDC subunits nor increased PDHA1 phosphorylation at S230, S293, or S300 (Extended Data Fig. 2i) upon KMT9 knockdown", they actually show in this panel a clear effect on S300 phosphorylation (but not on S230 nor on S293 phosphorylation). I don't think that this is problematic but they should not mislead the readers with such conclusion.

Version 2:

Reviewer comments:

Reviewer #1

(Remarks to the Author)

The authors have satisfactorily addressed my previous concerns. The current manuscript contains abundant data that support their conclusions, and is suitable for publication.

Reviewer #2

(Remarks to the Author)

The authors have fully addressed our comments.

Reviewer #3

(Remarks to the Author)

Reviewer #4

(Remarks to the Author)

This is a very nice piece of work uncovering a major role of the KMT9a methyltransferase in the control of pyruvate metabolism through DLAT monomethylation in prostate cancer development. The authors have answered my main concerns, in particular regarding the potential role of this KMT9a-DLAT crosstalk in the regulation of the nuclear pool of the PDC. Other technical issues and misunderstandings have been clarified. Although the authors do not yet provide a clear explanation about the tissue specific role of DLAT monomethylation in prostate cancer, I believe this Ms is now ready for publication at Nature Communications.

Minor comment

Line 228 : I guess the authors wanted to conclude that DLAT methylation does not change its lipoylation, right? The authors may want to correct this mistake.

REVIEWER COMMENTS

Reviewer #1 (Remarks to the Author):

The manuscript by Jia et al reported a specific role of mitochondrial KMT9 in prostate cancer cells for PDH activity, de novo lipogenesis, and cell proliferation. Mechanistically, they showed that this is achieved through mitochondrial KMT9 monomethylating DLAT at lysine 596. Evidence is provided using prostate cancer cell lines, a genetic mouse model, and human prostate specimens. This is an important study that reveals how a histone methyltransferase could regulate mitochondrial metabolism, which in turns influences prostate cancer progression. If these effects are truly specific to prostate cancer cells but not other tissues, it may represent a good therapeutic opportunity against prostate cancer by developing mitochondrial KMT9 inhibitors.

We thank reviewer #1 for the favorable judgment of our manuscript, constructive criticism, and valuable suggestions for revision.

Suggestions for revision:

Fig. 1: Is KMT9 present in mitochondria of primary prostate tissues? For example, the authors could test this using primary wildtype mouse prostate cells, and compare with expressions in primary tissues of other organs.

We appreciate the reviewer's valuable suggestion. Following this recommendation, we analyzed the expression of KMT9 α and KMT9 β in primary mouse prostate epithelial cells (new Extended Data Fig. 1g) and mouse tissues (prostate liver, spleen, kidney, heart, brain, bladder) (new Extended Data Fig. 1h). Mitochondrial KMT9 α was exclusively observed in the mouse prostate, while KMT9 β was also observed in mitochondria of spleen, brain and bladder. Since KMT9 α requires dimerization with KMT9 β for catalytic activity, active KMT9 can only form in mitochondria of mouse prostate tissue.

Changes to the main text of the revised manuscript describing the newly included figures can be found on page 5, lines 125-131.

Fig. 2e: No description of the properties of the human healthy mesenchyme or colon tissues or bladder and prostate cancer specimens were provided. How the tissues were acquired is also not mentioned.

Bladder and prostate cancer specimen were acquired by transurethral resection. For bladder cancer samples, the identification of cancerous tissue was based on a macroscopic assessment. For prostate cancer samples, the tissue condition was predefined by a previous cancer diagnosis. Procedures were executed in adherence to local guidelines of the Ethics Committee of the University of Freiburg (ETK: 266/14)

Healthy colon tissue as well as mesenchymal tissue specimens were procured during laparoscopic colon surgery. The identification and classification of non-neoplastic tissue were conducted through macroscopic evaluation in situ. For tissue samples derived from subjects with a prior diagnosis of colorectal adenocarcinoma, the specimens were specifically excised from regions proximal to the neoplastic lesion and were determined to be free of malignant tumor mass. Mesenchymal tissue was obtained from mesocolonic tissue in healthy regions. Procedures were executed in adherence to local guidelines of the Ethics Committee of the University of Freiburg (ETK: 21-1162_5).

We included the additional information in the revised manuscript (please see page 18, lines 565-577).

Fig. 3: Experiments were done mostly using prostate cancer cell lines. To show the effects are specific to prostate cancer, control cell lines from other cancer types would be desired. While HepG2 cells were used for Fig. 3n, other figure panels were prostate-only. For example, what will the effects be if MTS-KMT9 α is overexpressed in other cancer cell lines?

In the revised manuscript, we conducted a series of additional experiments according to the reviewer's valuable suggestions. First, we checked whether SW480 (colon cancer) and HT1376 (bladder cancer) cells are sensitive to DLAT depletion and observed similar sensitivity to DLAT knockdown compared to PCa cell lines (new Extended Data Fig. 3j-q). Importantly, proliferation defects of SW480 and HT1376 cells could be rescued with the methylation-defective mutant DLAT (K596R) showing that mechanisms other than methylation of K596 determine DLAT activity in these cell lines (new Extended Data Fig. 3m, 3p). Furthermore, contrary to prostate cancer cells, proliferation defects of SW480 and HT1376 cells upon KMT9 α depletion cannot be rescued with exogenous MTS-KMT9 α (new Extended Data Fig. 3ae, 3aj), whereas a partial rescue is observed in prostate cancer cells, e.g. DU-145 (Fig. 3e),

LNCaP and PC-3M (new Extended Data Fig. 3s, 3y). Similarly, free fatty acids and triglycerides are not reduced upon KMT9 α depletion in SW480 and HT1376 cells (new Extended Data Fig. 3ap, 3aq), which again is in contrast with PCa cells (new Extended Data Fig. 3an, 3ao). These control experiments further support the specific role of mitochondrial KMT9 α and DLAT methylation at K596 in prostate cancer cells.

Furthermore, we analyzed effects of KMT9 α depletion and inhibition of lipogenesis in four control cell lines (SW480, HT1376, MCF10A and Panc-1; new Extended Data Fig. 3bc-bf). Our analysis shows that combined KMT9 depletion and lipogenesis inhibition does not produce additive inhibitory effects on cell proliferation in these non-prostate cell lines. The data further support the PCa-specific nature of this regulatory mechanism.

Changes to the main text of the revised manuscript describing the newly included figures can be found on page 8, lines 237-241; page 9, lines 252-255; page 9, lines 266-271; and page 9, lines 283-287.

Fig. 3: To provide further evidence about cell proliferation, some xenografting experiments would be nice to add to the in vitro assays.

We acknowledge the value of xenograft experiments in cancer research. In this study, we validated our in vitro findings using a well-established and physiologically relevant knockout mouse model, which allowed us to examine the biological functions of KMT9 in a more native environment compared to xenografts. Since our mouse data are consistent with our in vitro observations, we think they clearly demonstrate the physiological relevance of our findings and are sufficient to support our conclusions within the scope of this manuscript.

Extended Fig. 3: While previous figures were done using the PC-3 cell line alongside DU145, this figure used LNCaP. Why not PC-3?

We thank the reviewer for this careful observation regarding cell line consistency. We corrected this inconsistency in the revised manuscript and included new data for PC-3M cells. Please, see new Extended Data Fig. 3c, g-i, x-ac, an, ao, av, aw, ay, ba, bh, bj. Our data demonstrate that the KMT9/PDC axis controls proliferation and lipogenesis also in PC-3M cells.

Changes to the main text of the revised manuscript describing the newly included figures can be found on page 8, lines 232-251; page 9, lines 263-266; page 9, lines 268-277; page 9, lines 282-285; and page 10, lines 287-290.

Fig. 4c: It is hard to believe that control prostate showed no Ki67 staining. Even for normal prostate homeostasis, a minority of epithelial cells should be Ki67-positive.

We thank the reviewer for this important observation. We indeed detect MKI67-positive epithelial cells, albeit at very low frequency (please see three selected images of control prostate tissue with MKI67-positive cells in Fig. 1a of the rebuttal letter along with quantification data from multiple independent samples). The observed low abundance of MKI67-positive cells in our control samples is in accordance with previous publications showing that epithelial cells in normal adult prostate maintain a remarkably low proliferation rate under homeostatic conditions (Blee, A. M. et al. (2018) *Clinical cancer research: an official journal of the American Association for Cancer Research* **24**, 4551-4565, doi:10.1158/1078-0432.ccr-18-0653; Traka, M. H. et al. (2010) *Molecular cancer* **9**, 189, doi:10.1186/1476-4598-9-189; Li, G. et al. (2020) *Cancer research* **80**, 4633-4643, doi:10.1158/0008-5472.can-20-0505). In fact, the control tissue image presented in Fig. 4c of the manuscript is more representative of the overall MKI67 staining pattern we typically observe in control prostates. We believe these additional data and analyses help to clarify the reliability of our MKI67 staining results.

Figure 1. Detection of MKI67-positive cells in control (Ctrl) mouse prostate tissue. a, fluorescent images showing MKI67-positive cells in selected fields of Ctrl mouse prostate tissue. Nuclei were counterstained with DAPI (gray). Scale bar: 50 μ m. b, Quantification of MKI67-positive cells across all analyzed fields from prostate tissues of indicated genotypes (n=6). Data represent mean \pm SD. ***P < 0.001 by Student's t-test.

Fig. 4: Does KMT9 conditional KO affect cell death rate in the tumor model?

To address the question, we performed TUNEL assays in Ctrl, *Pten/Smad4* KO, and *Pten/Smad4/KMT9 α* KO prostate tissues (new Extended Data Fig. 4b, c). Our data indicate that loss of KMT9 α does not significantly affect the cell death rate in this tumor model, which suggests that the observed phenotype is primarily driven by changes in cell proliferation rather than cell death.

Changes to the main text of the revised manuscript describing the newly included figures can be found on page 10, lines 312-315.

Reviewer #2 (Remarks to the Author):

In the authors' published paper (Metzger, Wang et al., Nat Struct Mol Biol 2019), they identified the 7- β -strand protein KMT9 as a novel histone H4K12 mono-methyltransferase and reported that KMT9 controls prostate cancer (PCa) cell proliferation in an enzymatic activity-dependent manner. There is a lack of correlation between H4K12me1 global levels and cellular phenotypes, implying other targets are involved in KMT9's function in PCa. In the current manuscript, the authors found KMT9 is localized in mitochondria specifically in PCa cells but not in other cancer cell types examined. Using carefully designed and extensive experiments, they demonstrated that mitochondrial KMT9 mono-methylates lysine 596 of the DLAT subunit in the pyruvate dehydrogenase complex (PDC). Further K596me1 on DLAT controls PDC activity, de novo lipogenesis and PCa cell proliferation. Finally, the authors showed that KMT9 plays an essential role in PDC activity, lipogenesis and tumor growth using a PCa mouse model. Overall, this manuscript is well prepared, and data support their conclusions. We have only a few comments.

We thank reviewers #2 and #3 for their favorable judgment of our manuscript, constructive criticism, and valuable suggestions for revision.

1. In Fig. 1 and Fig. 2, most experiments and conclusive data were derived from the PC-3M cell line. However, this cell line is not included in Fig. 3. Does the PC-3M cell line show any phenotype when DLAT K596me1 is disrupted?

We sincerely thank the reviewer for highlighting this important point about cell line consistency. To address this question and to correct inconsistencies in our manuscript, we included PC-3M cells in the new Extended Data Fig. 3c, g-i, x-ac, an, ao, av, aw, ay, ba, bh, bj. Our data show that the KMT9/PDC axis controls proliferation and lipogenesis also in PC-3M cells. These additional data further support our conclusion that this regulatory mechanism is conserved across multiple prostate cancer cell lines.

2. The conclusion related to Fig. 4 "mitochondrial KMT9 monomethylates DLAT at K596 to regulate PDC activity and lipogenesis in PCa in vivo" is overstated, because data in Fig. 4i-k are insufficient to support the role of DLAT K596me1 in PDC activity and lipogenesis in vivo.

We agree with the reviewers' point and changed the conclusion to "Together, these data show that the occurrence of mitochondrial KMT9 and mitochondrial DLAT K596me1 strongly correlates with KMT9-dependent differences in PDC activity, lipogenesis and prostate tumor growth in vivo." (Please see page 11, lines 331-333).

3. Fig. 4m-l: the y-axis can be stretched to better display differences between groups.

We adapted the scale of the figures to better display the differences between groups (new Fig. 4l,4m).

4. In the authors' previous paper, they showed that the percentage of PCa samples with nuclear KMT9 staining increased along with disease progression. Do authors also see elevated nuclear KMT9 levels in prostate tissue with higher Gleason grades?

We thank the reviewer for this insightful question connecting our current findings with our previous work. Indeed, in consistency with our previous data, we observe a statistically significant increase in nuclear KMT9 α (normalized to DAPI) for high Gleason grade tumors compared to normal or normal adjacent tissue (Extended Data Fig. 4d).

5. Please discuss why mitochondrial localization of KMT9 is only found in PCa.

The tissue-specific mitochondrial localization of KMT9 in prostate cells aligns with the emerging concept of tissue-specific regulation of mitochondrial functions. Several studies have demonstrated that mitochondrial properties and protein compositions vary significantly across different tissues, reflecting their adaptation to tissue-specific metabolic demands (Nunnari, J. & Suomalainen, A. (2012) *Cell* **148**, 1145-1159, doi:10.1016/j.cell.2012.02.035). For instance, the mitochondrial proteome shows substantial tissue-specific variations, with an estimated 20-30% of mitochondrial proteins being tissue-specific (Mootha, V. K. et al. (2003) *Cell* **115**, 629-640, doi:10.1016/s0092-8674(03)00926-7; Pagliarini, D. J. et al. (2008) *Cell* **134**, 112-123, doi:10.1016/j.cell.2008.06.016). In prostate cells, unique metabolic features have been well-documented, such as altered citrate metabolism and specific lipid synthesis pathways, which distinguish them from other cell types (Eidelman, E. et al. (2017) *Frontiers in oncology* **7**, 131, doi:10.3389/fonc.2017.00131; Bader, D. A. & McGuire, S. E. (2020) *Nat Rev Urol* **17**, 214-231, doi:10.1038/s41585-020-0288-x; Porporato, P. E. et al. (2018) *Cell Res* **28**, 265-280, doi:10.1038/cr.2017.155). These prostate-specific metabolic characteristics might necessitate specialized regulatory mechanisms, including tissue-specific mitochondrial protein localization. However, the exact mechanisms controlling the specific

localization of KMT9 in mitochondria of the prostate are currently unclear and will be elucidated in future investigations.

We adapted the discussion accordingly (please see page 12, lines 364-375).

6. Both KMT9 and PDC are found in nuclei, where DLAT K596me1 cannot be detected. Can authors comment on this?

We thank the reviewer for raising this mechanistic aspect. To address this question, we performed co-immunoprecipitation experiments with nuclear extracts. We did not observe an interaction between KMT9 and DLAT although both proteins are present in nuclei (new Extended Data Fig. 2h-k). This observation contrasts with complex formation in mitochondrial extracts (Fig. 1g). We hypothesize that yet unknown conditions may determine KMT9/DLAT interaction in mitochondria. We included these new results in the revised manuscript and adapted the main text accordingly (please see lines 198-201). Please also see the response to comment 4 of reviewer #4.

Kai Ge (NIH) and Guojia Xie (NIH)

Reviewer #3 (Remarks to the Author):

Reviewer #4 (Remarks to the Author):

Review Jia et al. Nature Communications 2024

In this Ms, Jia and collaborators describe a previously unknown effect of the methyl transferase KMT9 on the control of the pyruvate dehydrogenase complex (PDC). The authors made several original observations and identified DLAT (the E2 subunit of the PDC) as the target of mitochondrial KMT9. I believe this study provides interesting molecular insights on the regulation of the PDC, a key and central metabolic hub functioning at the crossroad of glycolysis and the TCA cycle. The data are globally very convincing and well presented, although several points need to be clarified.

We thank reviewer #4 for the favorable judgment of our manuscript, constructive criticism, and valuable suggestions for revision.

1- Could the authors try to estimate the size of the mitochondrial pool of KMT9 based on their immunoblots? Although their biochemical cell fractionation, in organello and fluorescence microscopy assays are consistent with KMT9 α/β mitochondrial localization, I think that electron microscopy data could further reinforce their conclusion.

We appreciate these constructive suggestions. We have now included detailed quantification of the mitochondrial KMT9 pool in the new Fig. 1a and Extended Data Fig. 1a, 1g. Our data show that approximately 5-10% of total KMT9 is present in mitochondria of the analyzed PCa cells.

The mitochondrial presence of KMT9 is supported by multiple independent experimental approaches in vitro and in vivo, including biochemical cell fractionation, in organello assays, and high-resolution confocal fluorescence microscopy. The convergence of these complementary techniques, each with distinct strengths and different underlying principles, provides robust validation of KMT9's mitochondrial localization. Given this comprehensive body of evidence establishing KMT9's mitochondrial presence, we believe that our current multi-method validation fully supports our conclusions. Thus, we feel that additional electron microscopy data would be beyond the scope of the manuscript.

2- Did the authors try to identify the domain involved in KMT9 mitochondrial import?

This is a very interesting point, but we currently only have preliminary observations concerning potential mechanisms of KMT9 mitochondrial localization. When analyzing the amino acid sequences of KMT9 α and KMT9 β , we only detected a putative mitochondrial targeting sequence (MTS) in the N-terminus of KMT9 β . After mutation of the putative MTS, KMT9 β retained its mitochondrial import capacity. These observations suggest that KMT9 mitochondrial import may be either achieved by a non-canonical import sequence or via alternative mechanisms, similar to what has been reported for other mitochondrial proteins (Chatterjee, A. et al. (2016) Cell **167**, 722-738, doi:10.1016/j.cell.2016.09.052; Lee, J. et al. (2005) J Biol Chem **280**, 40398-40401, doi:10.1074/jbc.C500140200; Li, M. X. et al. (2010) J Biol Chem **285**, 14871-14881, doi:10.1074/jbc.M109.069591; Marchenko, N. D. et al. (2000) J Biol Chem **275**, 16202-16212, doi:DOI 10.1074/jbc.275.21.16202; Mayer, A. et al. (1995) J Biol Chem **270**, 12390-12397, doi:DOI 10.1074/jbc.270.21.12390). At present, the mechanism of mitochondrial KMT9 import is elusive. Defining the mechanism of mitochondrial import represents a novel project beyond the scope of this manuscript.

3- Fig 1f. The data related to the identification of KMT9 interactors are very limited both in term of experimental design and description of the results. A table indicating the different interactors identified by LC-MSMS, integrating at least the number and sequence of the identified peptides, and the corresponding statistical analyses must be included. Furthermore, I think it would have been more logical to show the co-IP data confirming the KMT9-DLAT interaction (shown in Fig. 2b) in this panel.

We thank the reviewer for these helpful suggestions. We corrected the manuscript according to the reviewer's comments and included an Excel table with all relevant information (Extended Data Table 3). The Excel file contains three sheets showing all interactors of KMT9 α (sheet 'ALL'), providing information including sequences, quantities, and other parameters for all detected peptides derived from PDC subunits (sheet 'PDC subunits_peptide seq'), and the label-free quantification (LFQ) values for all detected PDC subunits along with p-values obtained from statistical analysis (sheet, 'PDC subunits_p value').

Additionally, following the reviewer's suggestion, we reorganized our figures by relocating the co-IP validation of KMT9-DLAT interaction (previously Fig. 2b) to new Fig. 1g.

4- One important question relates to the relevance of this mechanism of regulation of the PDC in the nucleus. This question is addressed in their Ms but I believe a few controls are missing. KMT9 was initially described as a nuclear chromatin remodeller. Since a nuclear pool of the PDC has been described by at least 2 independent groups (including one which highlighted its importance in histone acetylation in prostate cancer cells (Chen et al., Nat Genetics 2018)), I think that it is essential that the authors investigate in more details a potential contribution of this KMT9-PDC crosstalk in the nucleus. Although the authors have performed a complementation assay with a NLS-fused version of KMT9, I don't think that this is sufficient to prove that there is no impact of KMT9 on nuclear PDC. Indeed, despite the authors show that this NLS-fused ectopic KMT9 is needed to fully rescue the proliferation defect of KMT9-depleted cells, they don't demonstrate that this NLS-KMT9 can still target DLAT and thereby regulate its potential effects on histone acetylation in prostate cancer cells. The authors should demonstrate that the NLS-peptide is not interfering neither with KMT9 methylation activity on DLAT (using their in vitro assay) nor with DLAT interaction in these cells (by co-IP experiments). The absence of an effect of this NLS-KMT9 on nuclear PDC could also be confirmed by immunoblotting of histone acetylation, in particular on histone H3K9Ac that was linked to PDC activity in prostate cancer cells by Chen and collaborators. Complementary to data shown in Fig 2f, other experimental evidence that would rule out a contribution of this nuclear DLAT-KMT9 crosstalk could be provided by immunofluorescence experiments using their anti-K596me1 antibody to show that the staining is restricted to mitochondria and is completely excluded from the nucleus in prostate cancer cells. A partial answer to this question is provided by their nice in situ data shown in Fig. 4d where we see poor nuclear staining of DLAT K596me1 but I think that confocal microscopy analyses at high magnification in prostate cancer cells would provide a more definitive answer to this question.

We thank the reviewer for raising these important questions regarding nuclear PDC regulation. To comprehensively address these points, we performed a series of additional experiments. First, we checked by confocal laser scan microscopy for the presence of DLAT K596me1 in the nucleus. While we clearly observed methylated DLAT in mitochondria, there were no signals for methylated nuclear DLAT (new Extended Data Fig. 2g). Next, we performed co-immunoprecipitations using DLAT and endogenous KMT9 α or exogenous NLS-KMT9 α with anti-DLAT or anti-KMT9 α antibodies from nuclear cell extracts (new Extended Data Fig. 2h-k). In contrast to mitochondrial extracts (Fig. 1g), neither endogenous nor exogenous KMT9 α was co-immunoprecipitated (new Extended Data Fig. 2h-k). We hypothesize that the DLAT/KMT9 interaction may require an additional factor or condition present in mitochondria but not in the nucleus. In accordance with the lack of DLAT/KMT9 interaction in the nucleus, we found no changes in H3K9 acetylation upon KMT9 α knockdown or upon overexpression of NLS-KMT9 α (Extended Data Fig. 2l, m). Finally, we validated whether NLS-KMT9 α - in principle - can form a functional heterodimer with KMT9 β or whether the NLS interferes with function. For this purpose, we performed in vitro methylation assays with purified, recombinant NLS-KMT9 α /KMT9 β heterodimer and PDC isolated from PC-3M or DU145 cells. Our data clearly show that NLS-KMT9 α /KMT9 β is an active complex and that the NLS does not interfere with KMT9 function (new Extended Data Fig. 2n, o). In summary, these data do not provide any evidence for KMT9/PDC crosstalk in the nucleus and further strengthen our conclusions on the specific role of mitochondrial KMT9/PDC in PCa.

Changes to the main text of the revised manuscript describing the newly included figures can be found on page 7, lines 190-206.

5- I don't really understand why inhibition of KMT9 and ACLY would synergize if they act in the same pathway. The ACLY inhibitor and dKMT9 depletion should recapitulate the same effects except if they used doses that do not inhibit completely their respective target. If not too difficult in their environment, can they provide quantitative data such as stable isotope tracing data using ¹³C-labelled pyruvate) and comment on that?

We agree with Reviewer #4 that inhibition of KMT9 and ACLY cannot synergize when acting in the same pathway. We think the perception that we wanted to imply synergy is a misunderstanding. We actually wanted to check the possibility that acting in the same pathway may allow lower dosing of each treatment, which appears to be the case. We rephrased the correspond passage in the manuscript to avoid any misunderstanding (**please see page 9, lines 278-287**).

The reviewer's suggestion to perform stable isotope tracing experiments with ¹³C-labeled pyruvate would certainly provide additional quantitative data. However, our current combination of biochemical, cellular, and functional assays provides robust evidence for the proposed pathway. The extensive characterization includes direct measurement of PDC activity, acetyl-CoA levels, and cellular metabolic outputs, which together establish a clear mechanistic framework. Furthermore, the observed concordance between genetic (KMT9 depletion) and pharmacological (ACLY inhibition) approaches strongly supports our conclusions. While isotope tracing could offer

complementary information, we believe it would not fundamentally alter our main conclusions about the functional relationship between KMT9 and ACLY in this context. Furthermore, our facility does not provide the possibility to perform such experiments.

Is the partial rescue of proliferation provided by MTS-KMT9 (but not by the NLS-KMT9) in KMT9-depleted cells abrogated by the ACLY inhibitor? Is the proliferation defect of KMT9 cells rescued (at least partly) by addition of oleate?

We appreciate these insightful questions. To further validate our proposed mechanism, we performed the suggested experiments, and the results strongly support our model. As shown in new Extended Data Fig. 3bg-bj, the partial rescue of PCa cell proliferation by MTS-KMT9 α is indeed abrogated by SB204990, whereas there is no effect on the partial rescue by NLS-KMT9 α . Furthermore, the proliferation defect of PCa cells upon KMT9 α depletion is partially rescued by oleate (new Extended Data Fig. 3ax, 3ay). Together, the experiments further confirm the role of mitochondrial KMT9 in de novo lipogenesis in PCa.

Changes to the main text of the revised manuscript describing the newly included figures can be found on page 10, lines 287-290 and page 9, lines 275-277.

Minor comments:

- **Fig 1e: the authors should show LMNA immunoblots to demonstrate the purity of their mitochondrial fraction and the absence of contamination with nuclear material.**

We appreciate the reviewer's suggestion to further validate the purity of our mitochondrial preparations. In the revised manuscript, we provide immunoblots to demonstrate that our mitochondrial fractions are not contaminated with nuclear material (new Fig. 1e and Fig. 2 below).

Figure 2. The purity of mitochondrial fractions of the indicated cells lines was assessed by Western blot using an anti-LMNA antibody. Nuclear lysate from DU145 cells served as positive control.

- **Fig 2a: The authors should not indicate in the legend of this panel that they are detecting DLAT methylation per se with this IP-Western assay. Although this is likely the case, it is impossible to formally prove that the anti-pan methyl antibody recognizes DLAT but not another subunit of the immunoprecipitated PDC in this assay. The DLAT immunoblot only shows that DLAT is indeed present in their immunoprecipitated PDC complex. Only their MS assay and the experiments based on the K596 mutant demonstrate DLAT monomethylation by KMT9.**

We agree with the reviewer's point of view and adapted the results and figure legend of Fig. 2 accordingly. The modified text in the results section now reads: 'Our assays revealed methylation of a protein with the molecular weight corresponding to that of DLAT, which was abolished upon KMT9 α depletion (Fig. 2a)' (please see page 6, lines 170-172). The corrected figure legend now reads: 'Western blot analysis of PDC immunocaptured from mitochondrial fractions of siCtrl- or siKMT9 α -treated PC-3M cells using anti-pan-methyllysine antibody' (please see page 22, lines 708-709).

- **Extended Fig 2j. I think the authors should change their conclusion regarding potential crosstalks between KMT9 and other known regulatory mechanisms impinging on the PDC. Indeed, although they conclude that they found "no significant changes of the expression levels of PDC subunits nor increased PDHA1 phosphorylation at S230, S293, or S300 (Extended Data Fig. 2i) upon KMT9 knockdown", they actually show in this panel a clear effect on S300 phosphorylation (but not on S230 nor on S293 phosphorylation). I don't think that this is problematic but they should not mislead the readers with such conclusion.**

We appreciate the opportunity to clarify this point, since there seems to be a misunderstanding. The clear effect referee #4 is referring to (reduced PDHA1 S300 phosphorylation) is observed upon the knockdown of DLAT (new Extended Data Fig. 2s). Upon KMT9 α knockdown, we do not observe a clear effect on PDHA1 phosphorylation (potentially a small decrease in Ser293 phosphorylation) (new Extended Data Fig. 2r). However, to more accurately describe our observations and avoid misleading statements, we changed the corresponding paragraph to "As additional controls, we investigated whether DLAT methylation by KMT9 indirectly affected PDC function. KMT9 α

knockdown did not significantly affect the expression levels of PDC subunits (new Extended Data Fig. 2r). Furthermore, we checked for increased PDHA1 phosphorylation at S232, S293, or S300, which is known to inhibit PDC activity (Kolobova, E. et al. (2001) *Biochem J* **358**, 69-77, doi:Doi 10.1042/0264-6021:3580069) but observed only a small decrease in phospho-S293 and no significant changes for S232 and S300. However, upon DLAT knockdown, we noted a decrease in phosphorylation of S300, whereas phospho-S232 and -S293 levels were unchanged (new Extended Data Fig. 2s). Finally, exogenous DLAT and DLAT (K596R) were lipoylated to the same extent in DLAT-depleted cells (new Extended Data Fig. 2s), which shows that lipoylation, an indispensable posttranslational modification for PDC activity (Lacroix, M. et al. (2016) *Proc Natl Acad Sci U S A* **113**, 10998-11003, doi:10.1073/pnas.1602754113; Mathias, R. A. et al. (2014) *Cell* **159**, 1615-1625, doi:10.1016/j.cell.2014.11.046) is not indirectly affected by DLAT mutation." **(Please see page 8, lines 218-228).**

REVIEWER COMMENTS

Reviewer #1 (Remarks to the Author):

The authors have satisfactorily addressed my previous concerns. The current manuscript contains abundant data that support their conclusions, and is suitable for publication.

We thank the reviewer #1 for the positive evaluation of our manuscript.

Reviewer #2 (Remarks to the Author):

The authors have fully addressed our comments.

We thank the reviewer #2 for the positive evaluation of our manuscript.

Reviewer #3 (Remarks to the Author):

We thank the reviewer #3 for the positive evaluation of our manuscript.

Reviewer #4 (Remarks to the Author):

This is a very nice piece of work uncovering a major role of the KMT9a methyltransferase in the control of pyruvate metabolism through DLAT monomethylation in prostate cancer development. The authors have answered my main concerns, in particular regarding the potential role of this KMT9a-DLAT crosstalk in the regulation of the nuclear pool of the PDC. Other technical issues and misunderstandings have been clarified. Although the authors do not yet provide a clear explanation about the tissue specific role of DLAT monomethylation in prostate cancer, I believe this Ms is now ready for publication at Nature Communications.

We thank the reviewer #4 for the positive evaluation of our manuscript.

Minor comment

Line 228: I guess the authors wanted to conclude that DLAT methylation does not change its lipoylation, right? The authors may want to correct this mistake.

We thank the reviewer for pointing at a text passage that was not fully clear. In this experiment, our intention was to show that the K596R mutation, while preventing methylation at K596, does not indirectly affect DLAT lipoylation, a known critical posttranslational modification for PDC activity. By showing that both wild-type DLAT and DLAT (K596R) have comparable lipoylation levels, we confirm that the reduced PDC activity upon DLAT mutation is due to the loss of methylation at K596, rather than an indirect effect on lipoylation. We revised this section to make our reasoning clearer (Page 8, line 222-227).